# Tumor initiating cells induce Cxcr4-mediated infiltration of pro-tumoral macrophages into the brain

**Kelda Chia[1], Julie Mazzolini[1], Marina Mione[2], Dirk Sieger[1]\***

[1]Centre for Discovery Brain Sciences, University of Edinburgh, Edinburgh, United Kingdom; [2]Centre for Integrative Biology (CIBIO), University of Trento, Trento, Italy

**Abstract** It is now clear that microglia and macrophages are present in brain tumors, but whether or how they affect initiation and development of tumors is not known. Exploiting the advantages of the zebrafish (*Danio rerio*) model, we showed that macrophages and microglia respond immediately upon oncogene activation in the brain. Overexpression of human AKT1 within neural cells of larval zebrafish led to a significant increase in the macrophage and microglia populations. By using a combination of transgenic and mutant zebrafish lines, we showed that this increase was caused by the infiltration of peripheral macrophages into the brain mediated via Sdf1b-Cxcr4b signaling. Intriguingly, confocal live imaging reveals highly dynamic interactions between macrophages/microglia and pre-neoplastic cells, which do not result in phagocytosis of pre-neoplastic cells. Finally, depletion of macrophages and microglia resulted in a significant reduction of oncogenic cell proliferation. Thus, macrophages and microglia show tumor promoting functions already during the earliest stages of the developing tumor microenvironment.
DOI: https://doi.org/10.7554/eLife.31918.001

## Introduction

Microglia are the resident macrophages of the brain (for review see [*Kettenmann et al., 2011*; *Casano and Peri, 2015*]). Microglia are derived from primitive macrophages that invade the brain during development (*Ginhoux et al., 2010*; *Schulz et al., 2012*; *Herbomel et al., 2001*). Due to the neuronal environment, they finally differentiate to microglia and build up a resident population that is almost evenly dispersed throughout the central nervous system. Microglia fulfil a tremendous repertoire of functions in the brain. In addition to their classical immune functions, numerous studies have described roles for microglia in brain development, vessel patterning, synaptic pruning, the regulation of neuronal activity and even the influence of certain animal behaviors (*Casano and Peri, 2015*).

Despite the variety of beneficial functions in the brain, microglia can also act detrimentally during certain pathologies. Brain tumors represent probably the most severe example of harmful microglial functions. Microglia and infiltrating macrophages have been shown to infiltrate high-grade gliomas and can account for up to 30% of the tumoral mass (*Graeber et al., 2002*; *Badie and Schartner, 2001*; *Li and Graeber, 2012*; *Yang et al., 2010*; *Coniglio and Segall, 2013*). Instead of anti-tumoral activity, they display pro-tumoral functions and promote tumor growth. Macrophages and microglia have been shown to promote tumor cell proliferation and invasiveness, to modify the extracellular matrix, to induce angiogenesis and to induce an immunosuppressive environment promoting tumor growth (*Zhai et al., 2011*; *Zhang et al., 2012*; *Markovic et al., 2005*; *Komohara et al., 2008*; *Pyonteck et al., 2013*; *Wu et al., 2010*; *Hambardzumyan et al., 2016*; *Wang et al., 2013*; *Ellert-Miklaszewska et al., 2013*). Interestingly, these pro-tumoral functions seem to be independent of the tumor grade, as they have also been described for low-grade

**\*For correspondence:**
dirk.sieger@ed.ac.uk

**Competing interests:** The authors declare that no competing interests exist.

**eLife digest** Brain tumors can be aggressive, difficult to treat and are often incurable. Removing brain tumors by surgery can be challenging because the tumor cells infiltrate into the healthy tissue. Brain tumors grow in close physical contact with other cells, such as cells of the immune system. This includes cells called macrophages and microglia, which normally defend us against injuries and infections. However, instead of acting against the tumor as one might expect, macrophages and microglia actually support the growth of brain tumors.

It is not clear how and when during the development of a brain tumor the macrophages and microglia start helping the tumor cells to grow. Previous studies in this area have focused on these cell types found in advance brain tumors. Chia et al. have now looked at the earliest stages of tumor development, monitoring how macrophages and microglia respond to cancer cells.

To mimic human brain tumors, Chia et al. expressed a cancer-promoting version of a human protein in nerve cells of zebrafish larvae. These cells started behaving like early tumor cells. Live microscopy revealed that the tumor cells attract macrophages and microglia into their area, and that they also activate the immune cells. Biochemical experiments showed that the early tumor cells make and release a protein called Sdf1. Macrophages and microglia sense Sdf1 in the environment with a protein called Cxcr4 on their cell surfaces. When the gene for Cxcr4 was deleted in the zebrafish, the macrophages and microglia were not recruited into the developing tumors. When macrophages and microglia were depleted from the zebrafish larvae, the nerve cells with the mutant protein grew less well, supporting the idea that the immune cells enhance the development of early tumors.

A better understanding of how tumor cells and immune cells interact in the brain may help in the search for new anti-cancer drugs. Furthermore, the way in which macrophages and microglia are recruited to the tumor cells could be similar when tumors return after treatment. Future studies will test this hypothesis and, if it proves true, interfering with the macrophage and microglia response might delay the relapse of tumors.

DOI: https://doi.org/10.7554/eLife.31918.002

gliomas (*Daginakatte and Gutmann, 2007*; *Daginakatte et al., 2008*; *Simmons et al., 2011*). However, the earliest responses of macrophages and microglia to cancerous cells in the brain have not been addressed so far. It is not known when macrophages and microglia respond to oncogenic transformations in the brain and which signals are involved. Furthermore, we do not know how these initial responses physically appear. Are aberrant oncogenic cells initially removed by macrophages and microglia? Do pro-tumoural activities of macrophages and microglia develop only at later tumor stages? Or is the behavior of macrophages and microglia already co-opted during the early oncogenic stages in a way that they promote tumor growth? Understanding these early events is crucial as it is tempting to speculate that similar responses of macrophages and microglia occur during tumor recurrence. Thus, identifying the underlying mechanisms is a first step toward the development of new therapeutic strategies that target macrophages and microglia within brain tumors.

To address the early responses of macrophages and microglia to oncogenic alterations in the brain, we made use of the larval zebrafish model. We and others have already demonstrated that the zebrafish is a powerful tool to study microglia (*Peri and Nüsslein-Volhard, 2008*; *Sieger et al., 2012*; *Xu et al., 2016*; *Oosterhof et al., 2017*; *Svahn et al., 2013*; *Shiau et al., 2013*; *Shen et al., 2016*). Besides the genetic and optical advantages of the zebrafish larvae, the small size of the zebrafish brain allows imaging of the entire microglial network and its responses to pathological events. Furthermore, recent work has highlighted the suitability of the larval zebrafish model to investigate brain tumor growth (*Ju et al., 2014*; *Mayrhofer et al., 2017*; *Shin et al., 2012*; *Ju et al., 2015*; *Lal et al., 2012*; *Kitambi et al., 2014*; *Geiger et al., 2008*; *Lally et al., 2007*); *Yang et al., 2013*; *Jung et al., 2013*; *Hamilton et al. (2016)*; *Eden et al., 2015*; *Welker et al., 2016*). Here, we combined these advantages to study the responses of macrophages and microglia to early pre-neo-plastic cells in the brain. By overexpression of human dominant active AKT1 in neural cells, we induced abnormal cellular growth and morphology, increased proliferation and early differentiation, mimicking the earliest stages of brain tumor growth. Importantly, we detected an immediate

response of macrophages and microglia to these oncogenic events in the brain. This included a significant increase in macrophage and microglia cell numbers, which was caused by the infiltration of peripheral macrophages and mediated by Sdf1b released from oncogenic cells to activate the CXCR4b receptor on macrophages. Furthermore, macrophages and microglia showed highly dynamic cellular interactions with oncogenic cells. Finally, by reducing the number of macrophages and microglia, we showed that these cells actively promote proliferation of the pre-neoplastic cells in the brain.

## Results

### Expression of human AKT1 induces cellular transformation in the larval zebrafish brain

To investigate the response of macrophages and microglia to early pre-neoplastic cells, we overexpressed a dominant active version of the human AKT1 gene in the larval zebrafish brain (*Ramaswamy et al., 1999*). AKT1 is a downstream serine-threonine kinase in the RTK/PTEN/PI3K pathway which is highly activated in the majority of human glioblastomas (*McLendon and Cancer Genome Atlas Research Network, 2008*). Human AKT1 has been reported previously to induce malignant transformations in various neural cell types in zebrafish leading to glioma growth (*Mayrhofer et al., 2017*; *Jung et al., 2013*). To achieve expression in the nervous system, we expressed AKT1 under the neural-specific beta tubulin (NBT) promoter using a dominant active version of the LexPR transcriptional activator system (ΔLexPR) (*Emelyanov and Parinov, 2008*; *Mazaheri et al., 2014*). The NBT promoter is active in neural cells throughout zebrafish development (*Peri and Nüsslein-Volhard, 2008*). We either injected a lexOP:*AKT1*-lexOP:tagRFP construct into embryos of NBT:ΔLexPR:lexOP-pA (NBT:ΔLexPR) transgenic fish or co-injected an NBT:ΔlexPR-lexOP-pA driver plasmid together with a lexOP:*AKT1*-lexOP:tagRFP construct to achieve AKT1 expression (*Figure 1A*). In both scenarios, the NBT promoter drives expression of the transcriptional activator LexPR in neural cells which activates transcription of genes downstream the lexOP promoter sequence. These injections gave rise to mosaic expression of the oncogene within the larval nervous system (visualised through RFP expression of the construct) (*Figure 1*). During the course of development, AKT1 expression-induced morphological transformations resulting in larger cells with an abnormal morphology which was not observed upon injection of a lexOP:tagRFP control construct (*Figure 1B*). Immunohistochemistry using an anti-AKT1 antibody revealed strong expression of the human AKT1 protein upon lexOP:*AKT1*-lexOP:tagRFP construct injection (*Figure 1C*). To test for differentiation in AKT1 overexpressing cells, we performed immunohistochemistry for the differentiation marker Synaptophysin, which has been shown to be expressed in CNS tumors with neuronal differentiation as well as in neuroendocrine tumors (*Wiedenmann et al., 1986*). Indeed, expression of AKT1 led to an early onset of expression of Synaptophysin, which was not observed in controls (*Figure 1D*). To test for differentiation toward the glial lineage in AKT1-positive cells we stained for the glial cell marker GFAP. GFAP was neither detected in AKT1-expressing cells nor in RFP control cells at 8 dpf (*Figure 1E*). Intriguingly, oncogenic alterations have been previously shown to induce dedifferentiation of neuronal cells (*Friedmann-Morvinski et al., 2012*), thus we wondered if AKT1 overexpression is capable of inducing dedifferentiation in zebrafish neural cells. Interestingly, immunohistochemistry for the stem cell marker Sox2 revealed a subset of AKT1 cells which were positive for Sox2 while none of the RFP control cells appeared to be positive for Sox2 (*Figure 1F*). Furthermore, the number of proliferating AKT1-positive cells was significantly increased compared to control RFP cells (assessed by immunostaining for proliferating cell nuclear antigen (PCNA; *Figure 1G*). Finally, the oncogene expression led to significant impairment in survival with only 35.6% (62/174 larvae) surviving until 30 days post-fertilization (dpf) upon AKT1 overexpression compared to 90.8% (118/130 larvae) in controls (*Figure 1H*). Examination of 20 AKT1 expressing fish at 1 month of age showed persistent expression of the RFP-tag in the brain, with variable mosaicism (*Figure 1—figure supplement 1A–H*). Some of the RFP expressing cells showed neuronal features, including RFP-labeled axons and dendrites (*Figure 1—figure supplement 1D*), while 1/3 of the brains showed large clones of RFP-expressing cells, which grew deeply in the brain parenchyma, apparently without disrupting the brain architecture. Immunostaining of these areas with a panel of antisera (PCNA, AKT1, GFAP and Synaptophysin) showed disrupted cell architecture, with increased mitotic figures,

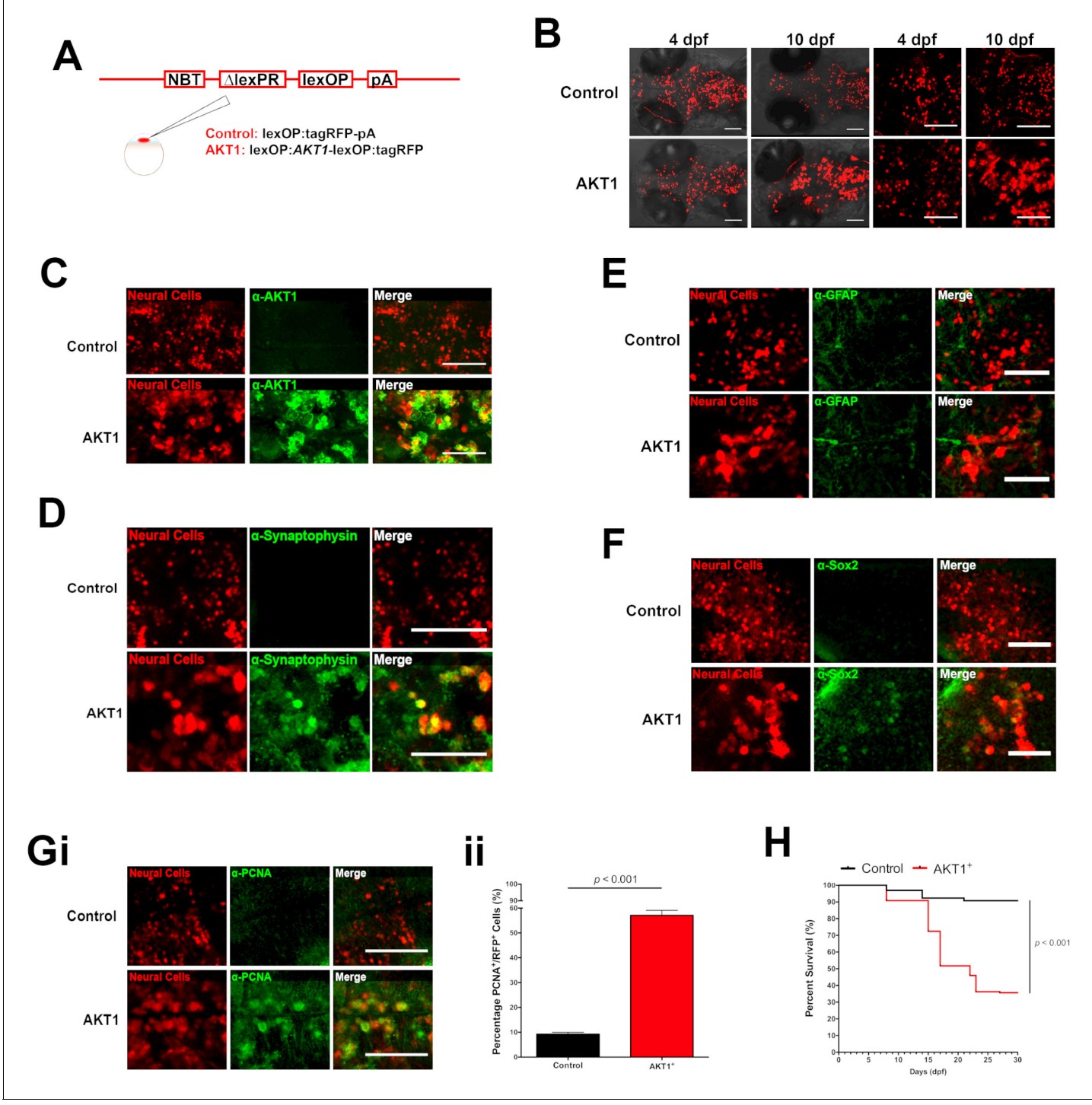

**Figure 1.** Human AKT1 induces transformation in the larval zebrafish brain. (**A**) To achieve expression in neural cells under the NBT promoter, the NBT:ΔLexPR-lexOP-pA zebrafish line was used. AKT1 expression was achieved through the injection of a lexOP:*AKT1*-lexOP:tagRFP into embryos at the one-cell stage. Control-RFP cells were obtained through the injection of a lexOP:tagRFP-pA. (**B**) In vivo imaging revealed early transformations and abnormal cellular morphology of AKT1-expressing cells in the brains of larval zebrafish from 4 dpf to 10 dpf. Representative confocal images of the larval zebrafish brain are shown. Upper panel: RFP control cells, lower panel: AKT1-expressing cells. (**C**) Immunohistochemistry revealed expression of the human AKT1 protein in AKT1-expressing cells, but not in control cells. Representative confocal images of the larval zebrafish brain at 8 dpf are shown. Upper panel: RFP control cells, lower panel: AKT1-expressing cells. (**D**) Immunohistochemistry revealed expression of the differentiation marker Synaptophysin in AKT1-expressing cells but not in control cells. Representative confocal images of the larval zebrafish brain at 8 dpf are shown. Upper panel: RFP control cells, lower panel: AKT1-expressing cells (**E**) Immunohistochemistry showed that neither AKT1-expressing cells nor control neural

*Figure 1 continued on next page*

*Figure 1 continued*

cells were positive for GFAP at 8 dpf. Representative confocal images of the larval zebrafish brain at 8 dpf are shown. Upper panel: RFP control cells, lower panel: AKT1-expressing cells. (F) Immunohistochemistry revealed that a subset of AKT1-expressing cells was positive for the stem cell marker Sox2 while neural control cells were negative. Representative confocal images of the larval zebrafish brain at 8 dpf are shown. Upper panel: RFP control cells, lower panel: AKT1-expressing cells. (Gi) Immunohistochemistry using the proliferation marker PCNA (proliferating cell nuclear antigen) revealed increased expression in AKT1-expressing cells compared to control cells. Representative confocal images of the larval zebrafish brain at 8 dpf are shown. Upper panel: RFP control cells, lower panel: AKT1 cells. (Gii) Quantification of the level of proliferation rates in RFP-positive neural cells in control and AKT1-positive fish, (Control: $9.2 \pm 0.75\%$, n = 13 larvae; Akt1: $57.1 \pm 2.03\%$, n = 17 larvae, $p<0.001$, N = 3) (H) Kaplan-Meier survival plot of control and AKT1 injected larvae over 30 days, n = 118/130, and 62/174 respectively. Error bars represent mean ± SEM. All images represent maximum intensity projections of confocal stacks. Images were captured using a Zeiss LSM710 confocal microscope with a 20X/NA 0.8 objective. Scale bars represent 100 µm.

DOI: https://doi.org/10.7554/eLife.31918.003

The following figure supplement is available for figure 1:

**Figure supplement 1.** Neural AKT1 expression induces brain tumors with mixed neuronal and glial components.

DOI: https://doi.org/10.7554/eLife.31918.004

as well as positive staining for AKT1, GFAP and Synaptophysin, suggesting abnormal growth and differentiation, suggestive of brain tumors with mixed neuronal and glial components (*Figure 1—figure supplement 1I–N*). The heterogeneous population of cells detected within these tumors is most probably a result of the early dedifferentiation events that we observed (*Figure 1F*). This is in line with previous observations in rodent models, showing that dedifferentiated cells are capable of generating a NSC or progenitor state which gives rise to heterogeneous populations observed in mature tumors (*Friedmann-Morvinski et al., 2012*).

In summary, these results confirm that human AKT1 has the capacity to induce pre-neoplastic alterations in the larval zebrafish brain.

## AKT1-expressing cells lead to increased microglia numbers and microglial activation

Microglia numbers have been shown to be increased in brain tumors (*Badie and Schartner, 2001*; *Graeber et al., 2002*; *Li and Graeber, 2012*; *Bowman et al., 2016*; *Chen et al., 2012*). However, it is unclear at which stage of tumor growth microglia numbers start to increase. To test if overexpression of AKT1 leads to an increase in microglial numbers, we used the 4C4 antibody that specifically labels microglia precursors and microglia in zebrafish (*Becker and Becker, 2001*; *Ohnmacht et al., 2016*) and quantified microglia numbers at 8 dpf (*Figure 2A,B*). As shown previously, AKT1 positive cells were observed to be highly proliferative at 8 dpf, with subsets of cells positive for Synaptophysin and others for Sox2, while GFAP-positive cells were not detected (*Figure 1D,E,F,G*). Interestingly, at this early time point we observed a significant increase in microglia numbers in AKT1-positive brains when compared to controls (*Figure 2B*). Furthermore, overexpression of AKT1 impacted on microglial morphology. Under normal physiological conditions, microglia show a highly ramified phenotype with several fine processes that are constantly extended and retracted to survey their environment (*Nimmerjahn et al., 2005*). In response to a pathological insult in the brain, the microglia retract their processes and assume an amoeboid morphology which reflects their activation (*Karperien et al., 2013*). This change in morphology leads to a change in the ratio of the microglia cell surface to microglia cell volume, which can be used as a read out for their activation (*Gyoneva et al., 2014*) (*Figure 2Ai–iii*). Based on these morphological criteria, we analyzed the microglia in AKT1 overexpressing brains and compared to control brains at 8 dpf (*Figure 2A,C*). This analysis showed a significantly higher percentage of amoeboid (activated) microglia in AKT1 overexpressing brains compared to control brains (*Figure 2C*).

In summary, these results show that early oncogenic events lead to an immediate response of the microglial network which is reflected in higher numbers and increased activation of microglia.

## Microglia show highly dynamic interactions with AKT1-positive cells

To analyze if microglia respond directly to AKT1 positive cells, we performed confocal live imaging using the mpeg1:EGFP transgenic zebrafish, in which all macrophages including microglia can be visualized and tracked (*Ellett et al., 2011*). Injection of lexOP:*AKT1*-lexOP:tagRFP or lexOP:tagRFP

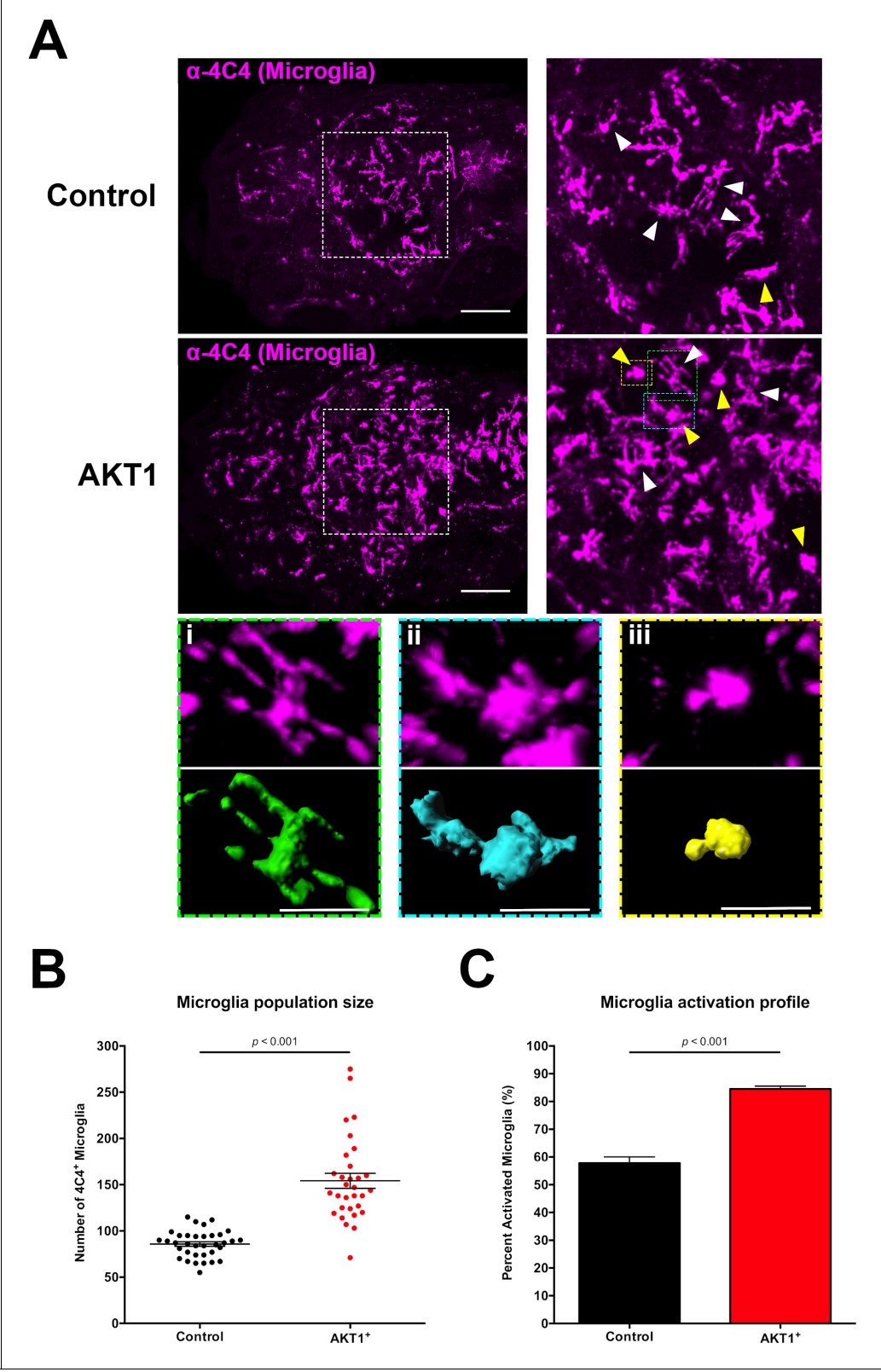

**Figure 2.** Induced transformation in AKT1-expressing cells leads to increased microglia numbers and microglial activation. (A) Immunohistochemistry using the microglia-specific antibody (α−4C4) showed increased microglia numbers and increased microglial activation upon overexpression of AKT1. Representative confocal images of the larval zebrafish brain are shown. Upper panels: upon control RFP expression, lower panels: upon AKT1 overexpression. White arrows indicate ramified microglia; Yellow arrows indicate amoeboid microglia. (Ai)-(Aiii) Higher magnifications of cells in the

*Figure 2 continued on next page*

*Figure 2 continued*

green (i), blue (ii) and yellow (iii) outlined areas. Upper panel shows 4C4 immunohistochemistry, lower panel shows segmented images using the surface tool in Imaris. (i) Represents a ramified cell (surface/volume ratio ~1), (ii) shows an activated cell (surface/volume ratio ~0.8) and (iii) shows a fully activated (amoeboid) cell (surface/volume ratio ~0.6) (B) Quantification of the numbers of microglia in the brain in control and following AKT1 overexpression (Control: 85.8 ± 2.45, n = 35 larvae; Akt1: 154.2 ± 8.15, n = 31 larvae, p<0.001, N = 3). (C) Quantification of the percentage activation within the microglial population in control and AKT1-positive fish (Control: 57.8 ± 2.26%, n = 10 larvae; AKT1: 84.5 ± 1.07%, n = 14 larvae, p<0.001, N = 2). Error bars represent mean ±SEM. Images were captured using a Zeiss LSM710 confocal microscope with a 20X/NA 0.8 objective. Scale bars represent 100 μm.

DOI: https://doi.org/10.7554/eLife.31918.005

constructs into oocytes of double transgenic mpeg1:EGFP/NBT:ΔLexPR fish allowed us to follow microglial responses to oncogenic cells or control cells in high temporal and spatial resolution. In control embryos, microglia showed the typical ramified morphology, and cell surface contacts between microglia and control RFP cells were only occasionally observed and of short durations (*Figure 3A*, *Video 1*, n = 7 samples analyzed). In line with our previous observations based on 4C4 immunohistochemistry (*Figure 2*), the density of microglia within AKT1-expressing brains was high and most microglial cells showed an amoeboid morphology (*Figure 3B*, *Video 2*, n = 9 samples analyzed). Interestingly, microglia were observed to stay within close vicinity to AKT1 positive cells throughout the duration of the experiment and most microglial cells were in direct surface contact with the oncogenic cells. These surface contacts were long lasting and consisted of different types. Microglia were observed to flatten their surfaces against the surfaces of AKT1-positive cells and then moving their surfaces around the surfaces of AKT1 cells for hours (*Figure 3B*, arrowheads, *Video 2*). Other microglial cells showed interactions with several AKT1-positive cells by constantly expanding and retracting their processes upon contact with surfaces of AKT1 positive cells (*Figure 3B*, arrows, *Video 2*). Importantly, many of these interactions were observed for several hours without any disruption. Thus, based on the persistence of these interactions it is unlikely that these were random contacts simply enforced by the high density of microglial and AKT1-positive cells. To address this further, we injected lower amounts of the lexOP:*AKT1*-lexOP:tagRFP construct into oocytes of double transgenic mpeg1:EGFP/NBT:ΔLexPR fish and screened for fish with lower numbers and isolated AKT1-positive cells. In these larvae, microglia were frequently seen to interact with AKT1-positive cells (*Figure 3C,D Video 3* and *4*, n = 6 samples analyzed). As observed previously, interactions lasted for several hours and consisted of different types from flattening of microglial surfaces against surfaces of AKT1-positive cells (*Figure 3C*) to constant extensions and retractions of processes toward AKT1 cells (*Figure 3D*). Thus, microglia are pro-actively interacting with AKT1-positive cells. Importantly, the observed interactions were not 'anti-tumoral' as we did not observe phagocytosis of AKT1-expressing cells.

Thus, changes induced by AKT1 overexpression stimulate direct cellular interactions with microglia.

## Macrophage infiltration in response to AKT1-induced transformation leads to increased numbers of 4C4 microglia

To address if the increased numbers of microglia in AKT1 positive brains were caused by proliferation of resident microglia, we performed co-immunostainings for microglia (4C4) and the proliferation marker PCNA. Interestingly, we did not detect a difference in the number of PCNA positive microglia when comparing AKT1-positive brains to control brains (not shown). Thus, we speculated that the presence of pre-neoplastic cells in the brain might cause an infiltration of peripheral macrophages, which then differentiate into 4C4-positive microglia-like cells. To address this question, we performed an L-plastin staining to identify all leukocytes in the brain (*Feng et al., 2010*). Through the co-labeling of L-plastin and 4C4, macrophages and neutrophils (L-plastin[+]/4C4[-]) can be distinguished from microglia (L-plastin[+]/4C4[+]). Indeed, we detected significantly higher numbers of L-plastin[+]/4C4[-] cells within the brain parenchyma upon AKT1 overexpression at 8 dpf, which resemble either newly infiltrated macrophages or neutrophils (*Figure 4A,Bi*). To test if neutrophils contributed to the higher number of L-plastin[+]/4C4[-] cells we performed AKT1 overexpression in transgenic mpx:GFP zebrafish in which neutrophils express GFP under the neutrophil-specific myeloperoxidase

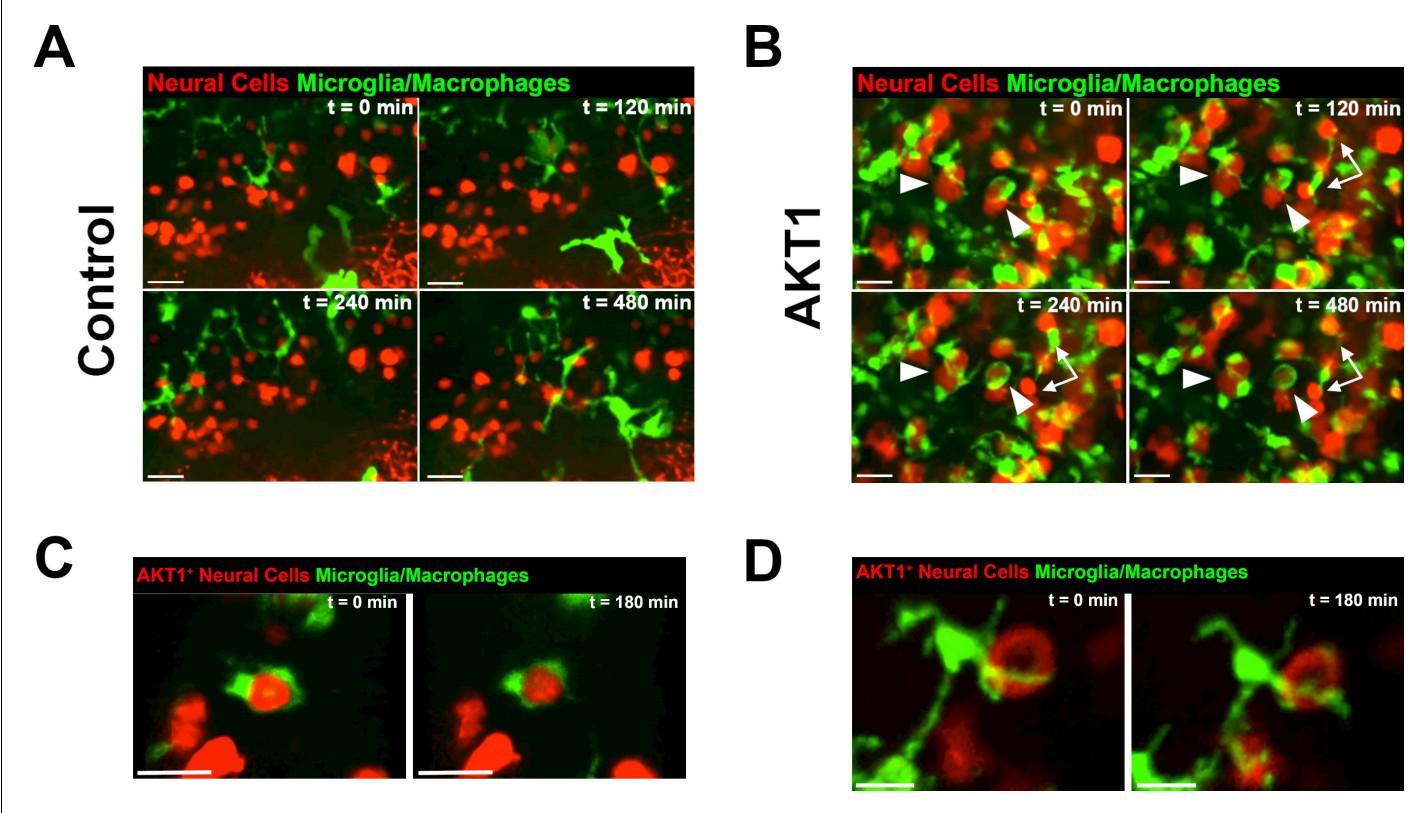

**Figure 3.** Microglia and macrophages show direct interactions with AKT1-expressing cells. The *Tg*(mpeg1:EGFP) line was outcrossed with the *Tg* (NBT:ΔLexPR-lexOP-pA) fish to create *Tg*(NBT:ΔLexPR-lexOP-pA; mpeg1:EGFP) double transgenic fish, in which macrophages/microglia express EGFP under the mpeg1 promoter. In vivo time-lapse imaging was carried out over a period of 8 hr (480 min) at 8 dpf to observe microglia/macrophage behavior within the brain parenchyma. (A) Microglia/macrophages were observed to behave physiologically in the presence of RFP-expressing control cells. Representative confocal images are shown, recording times indicated. See also *Video 1*. (B) Following AKT1 overexpression, microglia/ macrophages were observed to interact directly with the AKT1-expressing cells over long periods of time. Importantly, phagocytosis was not observed. Representative confocal images are shown, recording times indicated. See also *Video 2*. White arrows and arrowheads point at macrophages/microglia directly interacting with AKT1-expressing cells. (C), (D) Microglia interactions with AKT1-positive cells were targeted and also observed with isolated AKT1-positive cells. Representative confocal images are shown, recording times indicated. See also *Videos 3* and *4*. Images were captured using an Andor spinning disk confocal microscope with a 20X/NA 0.75 objective. Scale bars represent 30 μm.

DOI: https://doi.org/10.7554/eLife.31918.006

promoter (*Renshaw et al., 2006*). This experiment showed no infiltration of neutrophils upon AKT1 overexpression (data not shown), thus infiltrated L-plastin[+]/4C4[-] cells were solely macrophages.

In line with the previous results on increased microglial numbers, we detected a significant increase in the total amount of all L-plastin[+] cells following the overexpression of AKT1 compared to age-matched controls (*Figure 4A,Biii*). Within this population of L-plastin[+] cells, the majority of cells were positive for 4C4 (*Figure 4Bii*). As we did not detect proliferation of resident microglia, we hypothesized that infiltrated macrophages differentiated into microglia-like cells, leading to the higher numbers of 4C4-positive cells in AKT1-positive brains. If this hypothesis was true, then we should be able to detect an earlier time point when macrophages have just entered the brain but not differentiated to 4C4-positive cells yet. To test this, we performed L-plastin and 4C4 immunos-tainings at 3 dpf in AKT1-positive brains. Importantly, at 3 dpf we detected a 4.5-fold increase in the number of L-plastin[+]/4C4[-] cells in AKT1 positive brains compared to controls (*Figure 4Ci*). However, numbers for 4C4-positive microglia were similar to controls (*Figure 4Cii*). Thus, these L-plastin[+]/4C4[-] cells represented newly infiltrated macrophages. As numbers of 4C4[+] cells were only increased at later time points (*Figure 4Bii*) we conclude that these infiltrated macrophages differentiated into microglia like (4C4[+]) cells over time. To visualize these infiltration and differentiation events, we made use of a double transgenic model and overexpressed AKT1 in p2ry12:p2ry12-GFP/mpeg1:

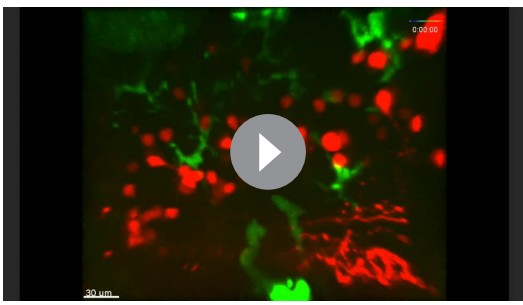 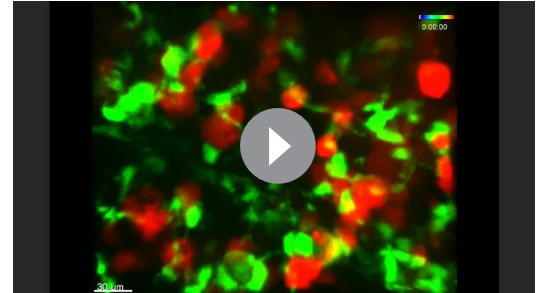

**Video 1.** Microglia/macrophage responses to control-RFP cells (REF to *Figure 2*). In vivo time-lapse movie showing representative microglia/macrophage (green) behavior in the presence of control cells (red). Images were acquired every 6 min over the duration of 8 hr (480 min) using an Andor spinning disk confocal microscope with a 20x/0.75 objective, n = 7 larvae analyzed. Scale bar represents 30 μm.
DOI: https://doi.org/10.7554/eLife.31918.007

**Video 2.** Microglia/macrophages display close interactions with AKT1+ cells (REF to *Figure 2*). In vivo time-lapse movie showing representative microglia/macrophage (green) behavior in the presence of AKT1+ cells (red). Compared to controls (REF *Video 1*), microglia/macrophages were observed to interact closely with AKT1+ cells over long periods of time. Images were acquired every 6 min over the duration of 8 hr (480 min) using an Andor spinning disk confocal microscope with a 20x/0.75 objective, n = 9 larvae analyzed. Scale bar represents 30 μm.
DOI: https://doi.org/10.7554/eLife.31918.008

mCherry zebrafish (*Ellett et al., 2011*; *Sieger et al., 2012*). In these zebrafish, all macrophages (including microglia) are positive for mCherry and microglia can be identified based on their additional P2ry12-GFP expression. To achieve AKT1 overexpression, we performed co-injections of the NBT:ΔLexPR driver plasmid and a lexOP:*AKT1*-lexOP:tagBFP expression plasmid. Intriguingly, we detected p2ry12-GFP$^-$/mCherry$^+$ macrophages at the dorsal periphery of AKT1-positive brains (*Figure 4—figure supplement 1*) at 5.5 dpf. Eventually, some of these macrophages infiltrated the brain parenchyma and started expressing GFP over time, showing their differentiation toward a p2ry12-GFP$^+$/mCherry$^+$ microglia-like cell (*Figure 4—figure supplement 1*, arrowheads, *Video 5*). These infiltration and differentiation events were not observed in control larvae (not shown), which is in line with previously published data showing that developmental colonization of the brain by primitive macrophages is happening before 4.5 dpf (*Casano et al., 2016*).

Importantly, similar observations have been made recently in a rodent glioma model where infiltrating monocytes take on a microglia-like identity (*Chen et al., 2017*).

In conclusion, these results show that early oncogenic events lead to a significant increase in the macrophage and microglia cell population in the brain.

## Cxcr4b signaling is required for the increase in macrophage and microglial numbers

We have shown that activation of AKT1 in neural cells leads to an increase in the macrophage and microglia cell population. To address the underlying mechanism, we focused on the chemokine receptor Cxcr4 as its role in the recruitment of tumor supportive macrophages has been shown previously (*Beider et al., 2014*; *Boimel et al., 2012*; *Hughes et al., 2015*; *Arnò et al., 2014*). To test a putative role for Cxcr4 in our model, we made use of the zebrafish *cxcr4b$^{-/-}$* mutant (*Haas and Gilmour, 2006*). To achieve overexpression of AKT1 in the *cxcr4b$^{-/-}$* mutant, we performed co-injections of the NBT:ΔLexPR driver plasmid and the lexOP:*AKT1*-lexOP:tagRFP expression plasmid into embryos of the mutant (*Figure 5A*). As in *cxcr4b* wild-type larvae, these injections resulted in a mosaic expression of the oncogene within the larval nervous system (*Figure 5B*). AKT1 expression induced morphological transformations resulting in larger cells with an abnormal morphology and expression of the human AKT1 protein in the *cxcr4b$^{-/-}$* mutant (*Figure 5B*). In line with this, we detected an early onset of expression of the differentiation marker Synaptophysin (*Figure 5C*). Thus,

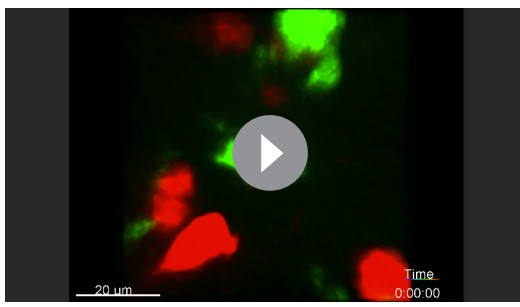
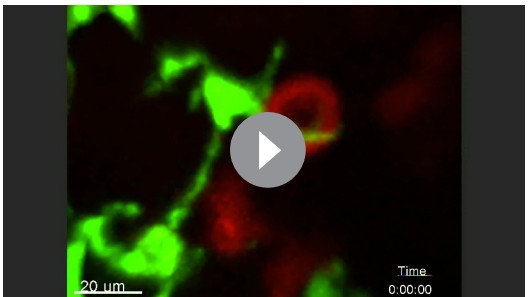

**Video 3.** Microglia/macrophages display close interactions with isolated AKT1[+] cells (REF to *Figure 2*). In vivo time-lapse movie showing representative microglia/macrophage (green) behavior in the presence of isolated AKT1[+] cells (red). Microglia/macrophages were observed to flatten their surfaces around the cellular surfaces of AKT1[+] cells over long periods of time. Images were acquired every 5 min over the duration of 3 hr (182 min) using an Andor spinning disk confocal microscope with a 20x/0.75 objective. Scale bar represents 20 µm.
DOI: https://doi.org/10.7554/eLife.31918.009

**Video 4.** Microglia/macrophages display constant interactions with isolated AKT1[+] cells (REF to *Figure 2*). In vivo time-lapse movie showing representative microglia/macrophage (green) behavior in the presence of isolated AKT1[+] cells (red). Microglia/macrophages were observed to constantly extend and retract processes toward cellular surfaces of AKT1[+] cells over long periods of time. Images were acquired every 6 min over the duration of 3 hr (180 min) using an Andor spinning disk confocal microscope with a 20x/0.75 objective. Scale bar represents 20 µm.
DOI: https://doi.org/10.7554/eLife.31918.010

overexpression of AKT1 in the *cxcr4b*[-/-] mutant induces alterations as observed in wild-type larvae. However, overexpression of AKT1 in the *cxcr4b*[-/-] mutant did not lead to an increase in microglia numbers compared to overexpression of AKT1 in wild-type larvae (*Figure 5D*). Notably, microglia numbers were similar in *cxcr4b*[-/-] controls and wild-type controls, showing that Cxcr4b signaling is not needed for the normal developmental population of the brain by microglia (*Figure 5D*). Thus, Cxcr4b signaling is either required for the infiltration of macrophages upon AKT1 overexpression or for their differentiation into 4C4-positive microglia-like cells. To address this question, we performed a co-labeling of L-plastin and 4C4 upon AKT1 overexpression in the *cxcr4b*[-/-] mutant. Intriguingly, the number of newly infiltrated macrophages (L-plastin[+]/4C4[-]) did not change upon AKT1 overexpression in the *cxcr4b*[-/-] mutant (*Figure 5Ei*). In line with this, we did not detect an increase in the total number of all L-plastin[+] cells upon AKT1 overexpression in the *cxcr4b*[-/-] mutant (*Figure 5Eii*). Importantly, *cxcr4b*[-/-] mutant zebrafish showed similar numbers of peripheral macrophages as wild-type zebrafish (*Figure 5—figure supplement 1*), thus the lack of infiltration of L-plastin[+] cells was not due to a general deficiency in peripheral macrophages. Furthermore, macrophages in *cxcr4b*[-/-] mutant embryos showed normal recruitment and function during mycobacterial infections in a recent study (*Torraca et al., 2017*). Thus, Cxcr4b signaling is specifically required for the infiltration of macrophages upon AKT1 overexpression in the brain.

To address if the lower numbers of microglia upon AKT1 overexpression in the *cxcr4b*[-/-] mutant impacted on the oncogenic cells, we analyzed their proliferation rates based on immunostainings for PCNA. Interestingly, while no observable impact on proliferating control-RFP cells was detected (*Figure 5F*), we detected an almost 50% reduced proliferation rate of AKT1-positive cells in the *cxcr4b*[-/-] mutant compared to the wild-type larvae following overexpression of AKT1 (*Figure 5F*).

These results show that Cxcr4b signaling is required for the infiltration of macrophages and subsequent increase in 4C4-positive microglia numbers in AKT1 positive brains. In addition, these data imply an impact of macrophages and microglia on the proliferation rate of pre-neoplastic cells. Alternatively, Cxcr4b might have a direct role in AKT1 positive cells and might be necessary for increased proliferation.

## Cxcr4b acts cell autonomously in macrophages and microglia

To test if Cxcr4b signaling acts cell autonomously in macrophages and microglia or AKT1 positive neural cells, we performed cell specific rescue experiments in the *cxcr4b*[-/-] mutant. To rescue Cxcr4b expression in macrophages/microglia, we made use of the macrophage/microglia-specific mpeg1

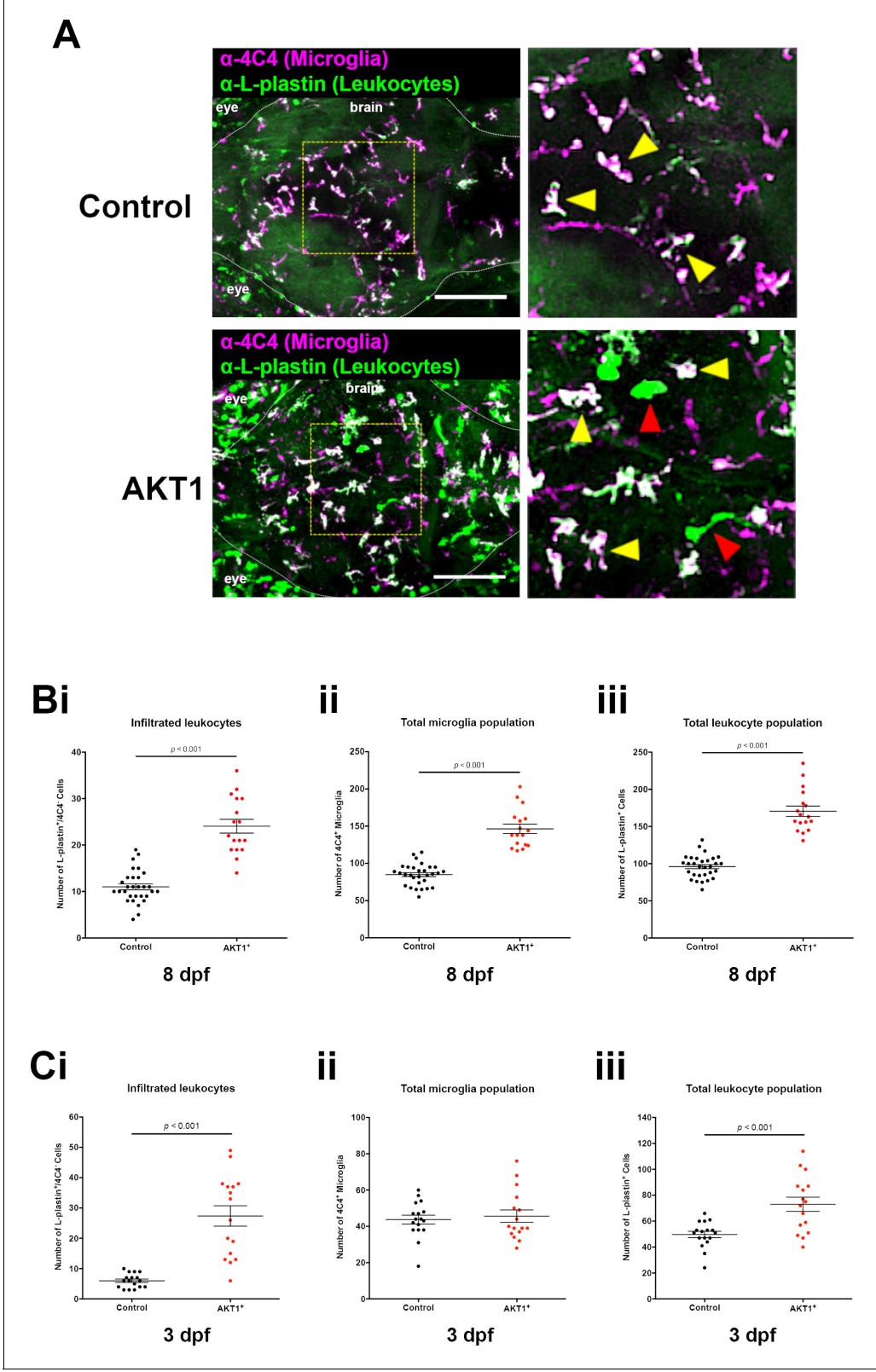

**Figure 4.** Infiltration of peripheral leukocytes into the brain parenchyma accounts for increased microglia numbers following AKT1 overexpression. (**A**) Immunohistochemistry carried out using the α−4C4 antibody for microglia and α-L-plastin antibody for leukocytes to distinguish microglial cells (L-plastin$^+$/4C4$^+$; indicated by yellow arrows) from newly infiltrating leukocytes (L-plastin$^+$/4C4$^-$; indicated by red arrows). This analysis revealed increased infiltration of leukocytes into the brains of AKT1-positive fish. Representative confocal images of the larval zebrafish brain are shown. Upper panels:
*Figure 4 continued on next page*

*Figure 4 continued*

upon control RFP expression, lower panels: upon AKT1 overexpression. The dotted white line demarcates the brain parenchyma (Bi) Quantification of the number of infiltrated leukocytes (L-plastin[+]/4C4[-]) into the brain parenchyma in control and AKT1-positive larvae at 8 dpf (AKT1: 24.1 ± 1.48, n = 17 larvae, and 11.0 ± 0.64, n = 30 larvae in age-matched controls, p<0.001, N = 3). (Bii) Quantification of the total 4C4[+] microglia population in control larvae and larvae following AKT1 overexpression at 8 dpf (Control: 85.0 ± 2.64, n = 30 larvae; AKT1: 146.5 ± 6.27, n = 17 larvae, N = 3). (Biii) Quantification of the total L-plastin[+] leukocyte population in control larvae and larvae following AKT1 overexpression at 8 dpf (Control: 96.0 ± 2.77, n = 30 larvae; AKT1: 170.5 ± 6.95, n = 17 larvae, p<0.001, N = 3). (Ci) Quantification of the number of infiltrated leukocytes (L-plastin[+]/4C4[-]) into the brain parenchyma in control and AKT1-positive larvae at 3 dpf (Control: 6.0 ± 0.56, n = 17 larvae; AKT1: 27.4 ± 3.38, n = 16 larvae, p<0.001, N = 2). (Cii) Quantification of the total 4C4[+] microglia population in control larvae and larvae following AKT1 overexpression at 3 dpf (Control: 43.7 ± 2.43, n = 17 larvae; AKT1: 45.6 ± 3.43, n = 16 larvae, p=1, N = 2). (Ciii) Quantification of the total L-plastin[+] leukocyte population in control larvae and larvae following AKT1 overexpression at 3 dpf, (Control: 49.7 ± 2.47, n = 17 larvae; AKT1: 73.0 ± 5.47, n = 16 larvae, p<0.001, N = 2). Error bars represent mean ±SEM. Images were captured using a Zeiss LSM710 confocal microscope with a 20X/NA 0.8 objective. Scale bars represent 100 μm.

DOI: https://doi.org/10.7554/eLife.31918.011

The following figure supplement is available for figure 4:

**Figure supplement 1.** Macrophages infiltrate AKT1-positive brains and start expressing *p2ry12*.

DOI: https://doi.org/10.7554/eLife.31918.012

promoter and injected a mpeg1:*cxcr4b* construct into embryos of *cxcr4b*[-/-] mutant fish in parallel to overexpression of AKT1 (*Figure 6A*). This led to a transient, mosaic expression of Cxcr4b on macrophages and microglia, thus reflecting a partial rescue (*Figure 6B*). To test the alternative that Cxcr4b expression is needed in the AKT1-positive neural cells, we performed a specific rescue of Cxcr4b in these cells. This was achieved by co-injection of a lexOP:*cxcr4b* plasmid in addition to a lexOP:*AKT1*-lexOP:tagRFP plasmid together with the NBT:ΔLexPR driver plasmid into the mutant background (*Figure 6A*). Again, this resulted in a transient, mosaic expression of Cxcr4b in oncogenic cells reflecting a partial rescue (*Figure 6B*). Upon cell-specific rescue of Cxcr4b in macrophages/microglia or AKT1-positive neural cells, microglia numbers in rescued larvae were analyzed. Interestingly, the rescue of Cxcr4b in macrophages/microglia led to an increase in 4C4 microglia numbers upon AKT1 overexpression in the *cxcr4b*[-/-] mutant (*Figure 6C*). On the contrary, the specific rescue of Cxcr4b expression in AKT1 positive neural cells did not lead to increased microglia numbers (*Figure 6C*). These results show that Cxcr4b function is needed cell autonomously in macrophages and microglia for an increase in microglial numbers induced by AKT1-overexpressing cells.

To further address a putative role of Cxcr4b in AKT1-positive neural cells, we analyzed the proliferation rate of AKT1-positive cells in the two specific rescue scenarios. Interestingly, while the Cxcr4b rescue in AKT1-positive neural cells did not increase their proliferation rates (*Figure 6D*), the specific rescue of Cxcr4b expression in macrophages and microglia led to a 70% increase in proliferation rates of AKT1-positive cells (*Figure 6D*). Thus, Cxcr4b signaling is not required in AKT1-positive neural cells to increase their proliferation rates. Importantly, these results imply a role for microglia in inducing proliferation in early pre-neoplastic cells.

## AKT1-positive cells express high levels of sdf1b

The Cxcr4b receptor can be activated by its cognate ligands Sdf1a (Cxcl12a) and Sdf1b (Cxcl12b) and the noncognate ligand macrophage migration inhibitory factor (Mif). To address which of these ligands is responsible for Cxcr4b activation upon AKT1 overexpression, we isolated and sorted AKT1-RFP-positive cells and control cells from larval brains by FACS and performed qPCR

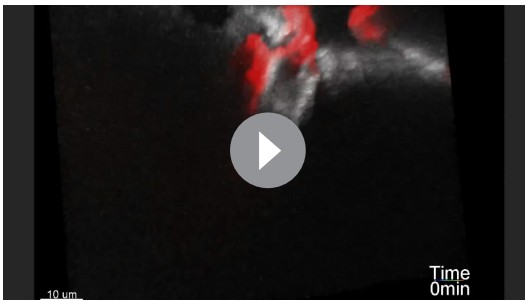

**Video 5.** Macrophages start expressing *p2ry12* upon infiltration into AKT1-positive brains. In vivo time-lapse movie showing macrophage (red) infiltration and activation of *p2ry12* expression (white) in AKT1-positive brains. Macrophages (red) were observed at the dorsal periphery infiltrating into the brain parenchyma. Immediately upon infiltration macrophages started expressing *p2ry12* (white). Images were acquired every 6 min over the duration of 2 hr (126 min) using an Andor spinning disk confocal microscope with a 20x/0.75 objective. Scale bar represents 10 μm.

DOI: https://doi.org/10.7554/eLife.31918.013

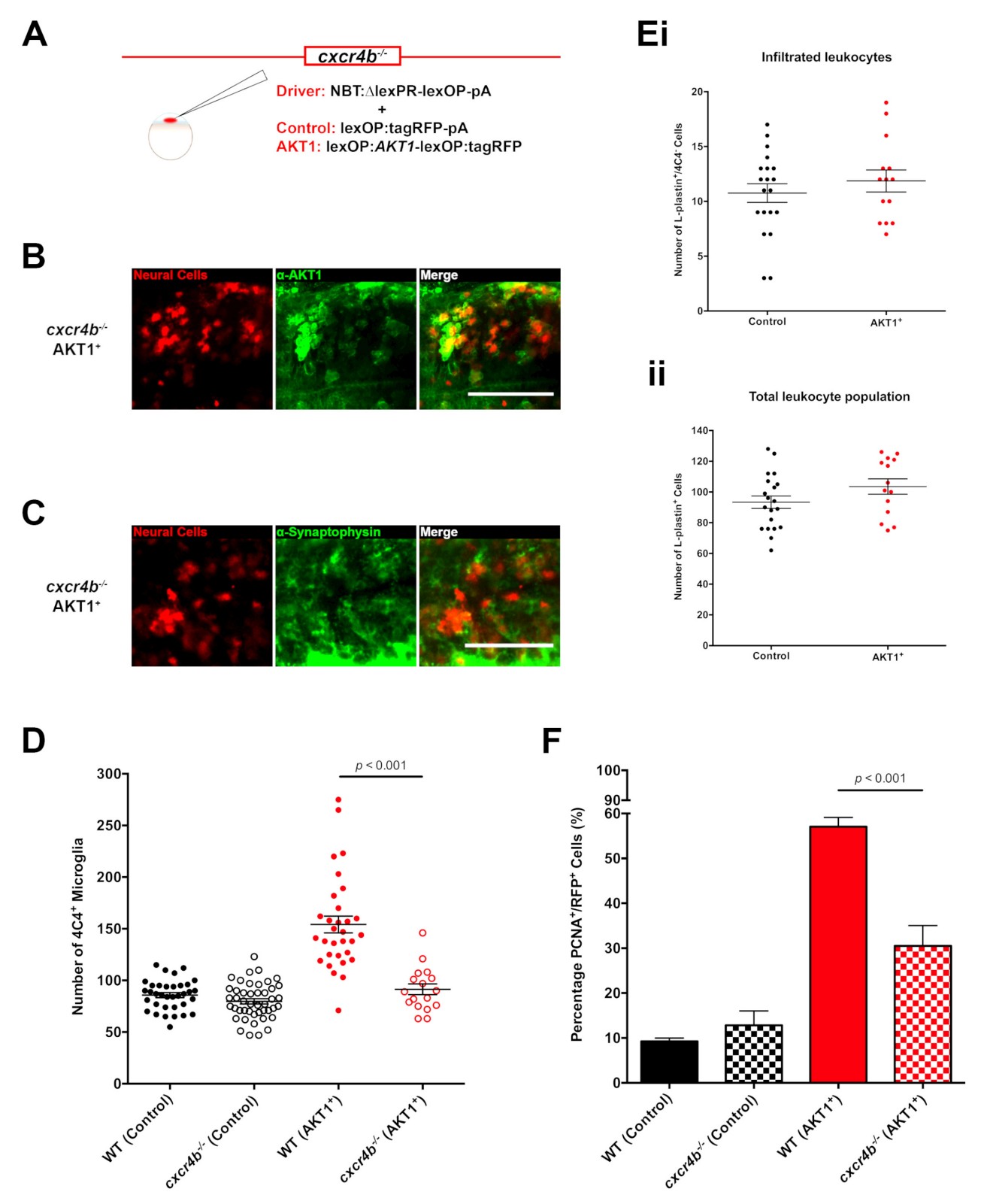

**Figure 5.** Cxcr4b signaling is required for the increase in microglia numbers following AKT1 overexpression. (**A**) To achieve expression in neural cells in the *cxcr4b*[-/-] mutant zebrafish line, a driver plasmid containing the NBT promoter (NBT:ΔlexPR-lexOP-pA) was co-injected together with either a lexOP: *AKT1*-lexOP:tagRFP plasmid to induce AKT1 expression or together with a lexOP:tagRFP-pA plasmid to achieve control RFP expression. Immunohistochemistry expression of (**B**) the human AKT1 protein and (**C**) Synaptophysin in the AKT1-expressing cells in the *cxcr4b*[-/-] mutant fish at 8

*Figure 5 continued on next page*

*Figure 5 continued*

dpf. (D) Quantification of the number of microglia in wild-type (WT)(*cxcr4b$^{+/+}$*) controls, *cxcr4b$^{-/-}$* controls, and following AKT1 overexpression in WT larvae and *cxcr4b$^{-/-}$* fish at 8 dpf (WT AKT1: 154.2 ± 8.15, n = 31 larvae; *cxcr4b$^{-/-}$* AKT1: 91.4 ± 5.22, n = 17 larvae, p<0.001, N = 3). (Ei) Quantification of the number of infiltrated leukocytes (L-plastin$^{+}$/4C4$^{-}$) into the brain parenchyma in control and AKT1-positive *cxcr4b$^{-/-}$* fish at 8 dpf (*cxcr4b$^{-/-}$* Control: 10.8 ± 0.85, n = 20 larvae; *cxcr4b$^{-/-}$* AKT1: 11.9 ± 1.00, n = 14 larvae, p=0.408 (n.s.), N = 3). (Eii) Quantification of the total L-plastin$^{+}$ leukocyte population in control *cxcr4b$^{-/-}$* larvae and following AKT1 overexpression at 8 dpf (*cxcr4b$^{-/-}$* Control: 93.4 ± 4.05, n = 20 larvae; *cxcr4b$^{-/-}$* AKT1: 103.5 ± 5.00, n = 14 larvae, p=0.122 (n.s.), N = 3). (F) Quantification of the level of proliferation rates in control-RFP and AKT1-expressing cells in WT and *cxcr4b$^{-/-}$* fish at 8 dpf (WT AKT1: 57.1 ± 2.03%, n = 17 larvae, *cxcr4b$^{-/-}$* AKT1: 30.5 ± 4.53%, n = 5 larvae, p<0.001, N = 2). Error bars represent mean ±SEM. Images were captured using a Zeiss LSM710 confocal microscope with a 20X/NA 0.8 objective. Scale bars represent 100 μm.
DOI: https://doi.org/10.7554/eLife.31918.014
The following figure supplement is available for figure 5:

**Figure supplement 1.** Cxcr4b$^{-/-}$ zebrafish larvae show a normal distribution of macrophages.
DOI: https://doi.org/10.7554/eLife.31918.015

for the respective ligands. While *sdf1a* expression levels did not show differences to control cells we detected a three-fold increase in *mif* expression levels and an over 10-fold increase in *sdf1b* expression levels in AKT1-positive neural cells compared to control cells (*Figure 7A*). To further test a putative role for Sdf1b in Cxcr4b induced microglial population increase, we performed Sdf1b overexpression in larval brains in the absence of AKT1 overexpression. Injection of a lexOP:*sdf1b*-lexOP:tagRFP construct into NBT:ΔLexPR transgenic fish led to significantly increased microglia numbers compared to controls (*Figure 7B*). Thus, higher levels of Sdf1b are sufficient to induce an increase in microglia numbers. To test if Sdf1b activity was solely mediated via Cxcr4b activation, we performed Sdf1b overexpression in the *cxcr4b$^{-/-}$* mutant and quantified microglia numbers. Indeed, in the *cxcr4b$^{-/-}$* mutant background, Sdf1b overexpression did not impact on microglia numbers (*Figure 7B*). Thus, high levels of Sdf1b produced by AKT1 positive cells lead to increased microglia numbers via Cxcr4b activation.

## Macrophages and microglia induce proliferation of AKT1-positive cells

We have shown that Cxcr4b signaling functions cell autonomously in macrophages and microglia during early stages of AKT1 activation in the brain. To further assess the impact of macrophages and microglia on AKT1-positive cells, we decided to inhibit microglial function independently of Cxcr4b signaling. We treated AKT1 overexpressing larvae with the immunosuppressant Dexamethasone (DEX) which has previously been shown to inhibit macrophages and microglia upon spinal cord injury and brain inflammation in zebrafish (*Ohnmacht et al., 2016*; *Kyritsis et al., 2012*). Furthermore, to specifically deplete macrophages and microglia, we treated AKT1-expressing larvae with the CSF-1R inhibitor Ki20227 that has previously been shown to deplete macrophages and to reduce tumor growth in a rodent melanoma model (*Tham et al., 2015*; *Ohno et al., 2006*). Treatment with DEX or Ki20227 had a direct impact on microglia in control and AKT1-positive larvae, leading to significantly reduced microglia numbers in controls and AKT1 larvae, respectively (*Figure 8Ai*). Furthermore, while proliferation rates of neural cells remained constant in controls (*Figure 8Aii*), DEX and Ki20227-treated AKT1-positive larvae showed a more than 50% reduction in the number of proliferating AKT1 cells compared to DMSO controls (*Figure 8Aii*).

Finally, to directly address the impact of macrophages and microglia on AKT1-positive cells without any pharmacological interference, we made use of the zebrafish *irf8$^{-/-}$* mutant (*Shiau et al., 2015*). Interferon regulatory factor 8 (Irf8) is vital for macrophage development in mammals and in teleosts. The *irf8* null mutant (*irf8$^{-/-}$*) zebrafish was characterized to lack macrophages up to around 6 dpf, with recovery from 7 dpf while microglia were absent in the brain until 31 dpf (*Shiau et al., 2015*). In line with published observations, we did not detect 4C4 positive microglial cells in *irf8$^{-/-}$* larvae at 8dpf (*Figure 8Bi*). Interestingly, upon AKT1 overexpression in *irf8$^{-/-}$* larvae, we detected a small population of 4C4-positive cells (*Figure 8Bi*). These cells were probably derived from the population of macrophages that recover in *irf8$^{-/-}$* larvae from 7 dpf, which then infiltrated into the brain due to AKT1 overexpression and differentiated into 4C4$^{+}$ cells. Importantly, the number of 4C4-positive cells was 80% lower in *irf8$^{-/-}$* larvae compared to wild-type larvae upon AKT1 overexpression (*Figure 8Bi*), which allowed us to address the impact on the proliferation of AKT1-positive cells. While proliferation rates of neural cells remained constant in *irf8$^{-/-}$* controls (*Figure 8Bii*), we

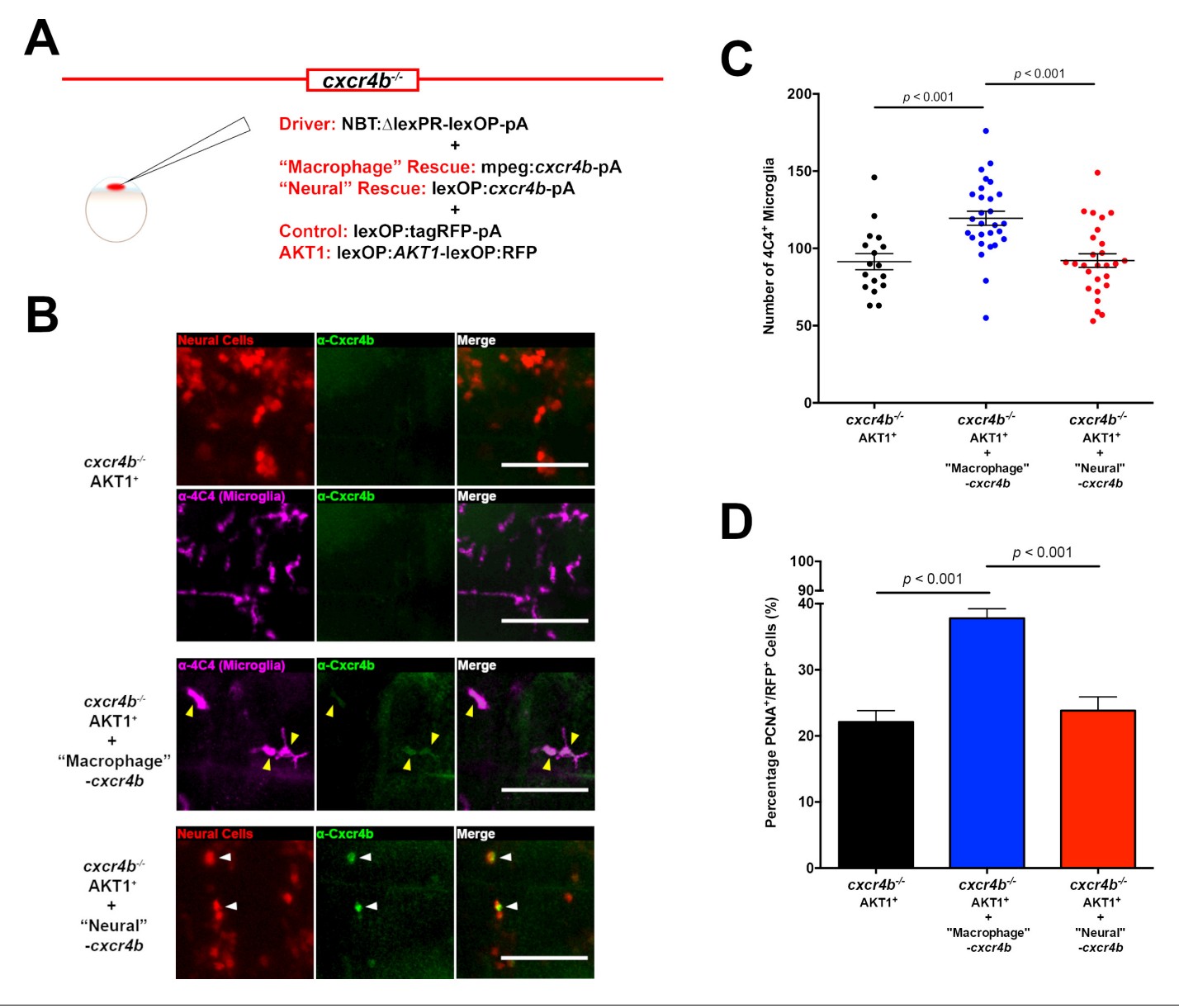

**Figure 6.** Cxcr4b signaling in macrophages is required for the increase in microglia numbers upon AKT1 transformation in the brain. (**A**) To rescue Cxcr4b expression in macrophages or neural cells in the *cxcr4b*[-/-] mutant fish, a cell-specific rescue construct was injected in addition to the NBT driver and lexOP:tagRFP-pA/lexOP:*AKT1*-lexOP:RFP constructs. Cxcr4b expression was recovered in macrophages through the mpeg1 promoter ('macrophage' rescue) via a mpeg1:*cxcr4b*-pA construct. The expression of cxcr4b in neural cells ('Neural' rescue) was rescued through the lexOP: *cxcr4b*-pA construct. (**B**) Immunohistochemistry using a Cxcr4b antibody showed that Cxcr4b expression is completely absent in microglia and neural cells in the *cxcr4b*[-/-] fish (upper panels). Cxcr4b expression is partially recovered following the respective rescue conditions (lower panels). (**C**) Quantification of the number of microglia in *cxcr4b*[-/-] mutants overexpressing AKT1 before, and after Cxcr4b rescue in macrophages ('Macrophage' rescue: 119.5 ± 4.59, n = 28 vs *cxcr4b*[-/-] Akt1: 91.4 ± 5.22, n = 17; p<0.001, N = 39) and in neural cells ('Neural' rescue: 92.2 ± 4.40, n = 27, p=1 (n.s.), N = 3). (**D**) Quantification of the level of proliferation of AKT1-expressing cells in *cxcr4b*[-/-] mutants overexpressing AKT1 before, and after Cxcr4b rescue in macrophages ('Macrophage' rescue: 37.8 ± 1.45%, n = 45 larvae, vs *cxcr4b*[-/-] AKT1: 22.1 ± 1.73%, n = 27 larvae p<0.001, N = 3) and in neural cells ('Neural' rescue: 23.8 ± 2.08%, n = 22 larvae, vs *cxcr4b*[-/-] AKT1: 22.1 ± 1.73%, n = 27 larvae, p=1.00 (n.s.), N = 3). Error bars represent mean ±SEM. Images were captured using a Zeiss LSM710 confocal microscope with a 20X/NA 0.8 objective. Scale bars represent 100 µm.

DOI: https://doi.org/10.7554/eLife.31918.016

detected an almost 60% reduction in proliferation of AKT1-positive cells compared to AKT1 positive cells in wild-type larvae (*Figure 8Bii*).

In summary, these experiments show that macrophages and microglia promote proliferation of AKT1-positive cells from the earliest stages of brain tumor growth.

## Discussion

In this study, we addressed the responses of macrophages and microglia to the earliest events of brain tumor growth. Several elegant studies have shown that macrophages and microglia infiltrate brain tumors and promote their growth (for review see (*Hambardzumyan et al., 2016*)). However, so far it was unclear when myeloid cells first respond to oncogenic events in the brain. We made use of the larval zebrafish model to address the earliest stages of tumor induction due to activation of oncogenes. By overexpressing a dominant active version of the human *AKT1* gene, we induced cellular alterations in the larval zebrafish brain. These were reflected in abnormal cellular morphology, increased proliferation and an early onset of Synaptophysin expression. Intriguingly, we detected an immediate response of macrophages and microglia to AKT1-induced cellular alterations. By using a combination of mutant and transgenic zebrafish lines, as well as immunohistochemistry for macrophages and microglia, we showed an immediate increase in the population of macrophage and microglia cells. This increase was the result of infiltrating macrophages that differentiated into microglia-like cells as observed by expression of the 4C4 antigen. These results are in line with recent data from rodent models investigating the contribution of macrophages and microglia to the myeloid cell population within gliomas. Based on multiple models of murine brain malignancy and genetic lineage tracing, Bowman et al. showed that infiltrating macrophages are abundant in primary and metastatic brain tumors (*Bowman et al., 2016*). A further study using genetically engineered mouse models to distinguish monocytes/macrophages from microglia showed that 85% of the tumor-associated macrophages (TAMs) within the glioma were infiltrated monocytes/macrophages (*Chen et al., 2017*). Furthermore, the authors concluded that infiltrating monocytes transitioned to macrophages and microglia-like cells (*Chen et al., 2017*). The increased number of 4C4 positive cells detected in our model might reflect these microglia-like cells. This is further supported by our live imaging experiments showing that infiltrating macrophages start expressing *p2ry12*, which has been shown to be microglia specific across different species (*Crotti and Ransohoff, 2016*). However, further markers will be required to test if these cells are microglia-like or fully differentiated microglial cells.

Interestingly, while the CCL2/CCR2 signaling axis seems to be the main attractant for monocytes at the later stages of glioma growth, we show that Sdf1b/Cxcr4b signaling is responsible for macrophage infiltration during the initial stages of oncogene activation in the brain. Importantly, a role for SDF1/CXCR4 signaling in macrophage attraction has been described before in rodent models of breast cancer and tumor relapse after chemotherapy (*Hughes et al., 2015*; *Boimel et al., 2012*). Our data show that the pre-neoplastic AKT1 cells produce Sdf1b, which is in line with previous studies showing that AKT1 is involved in activating the expression of SDF1 (*Furue et al., 2017*; *Conley-LaComb et al., 2013*). Other studies revealed an AKT-mediated induction of the transcription factor specificity protein 1 (Sp1) and Sp1-binding sites were shown to be functional within the human SDF1 promoter (*Gómez-Villafuertes et al., 2015*; *Pore et al., 2004*; *García-Moruja et al., 2005*). Thus, AKT1-induced Sdf1b expression might be mediated via Sp1.

Furthermore, we show that Sdf1b produced by the pre-neoplastic cells acts specifically on macrophages and microglia but is not required in an autocrine fashion during these early stages. By using a zebrafish mutant for *cxcr4b*, we showed that Cxcr4b signaling is required for macrophage infiltration upon AKT1 expression in the brain. Only the specific rescue of Cxcr4b in macrophages and microglia led to increased microglial numbers in the mutant background while a specific rescue in oncogenic cells did not alter the phenotype. Importantly, we also detected a significant decrease in the number of proliferating AKT1 cells in the *cxcr4b*$^{-/-}$ mutant background, which could not be rescued by Cxcr4b expression in these cells. Thus, Sdf1b/Cxcr4b signaling is not required for proliferation of AKT1-expressing cells in an autocrine fashion during initial stages of oncogenic activation. This might highlight another difference between the early and late stages of tumor growth, as autocrine SDF1/CXCR4 signaling has been shown to be involved in proliferation of various glioma cell lines (for review see (*Gagliardi et al., 2014*)).

Intriguingly, rescuing Cxcr4b expression in macrophages and microglia in the *cxcr4b*$^{-/-}$ mutant background led to a significant increase in proliferation of AKT1-positive cells, suggesting a role for microglia in promoting proliferation of pre-neoplastic cells. To further address this, we depleted

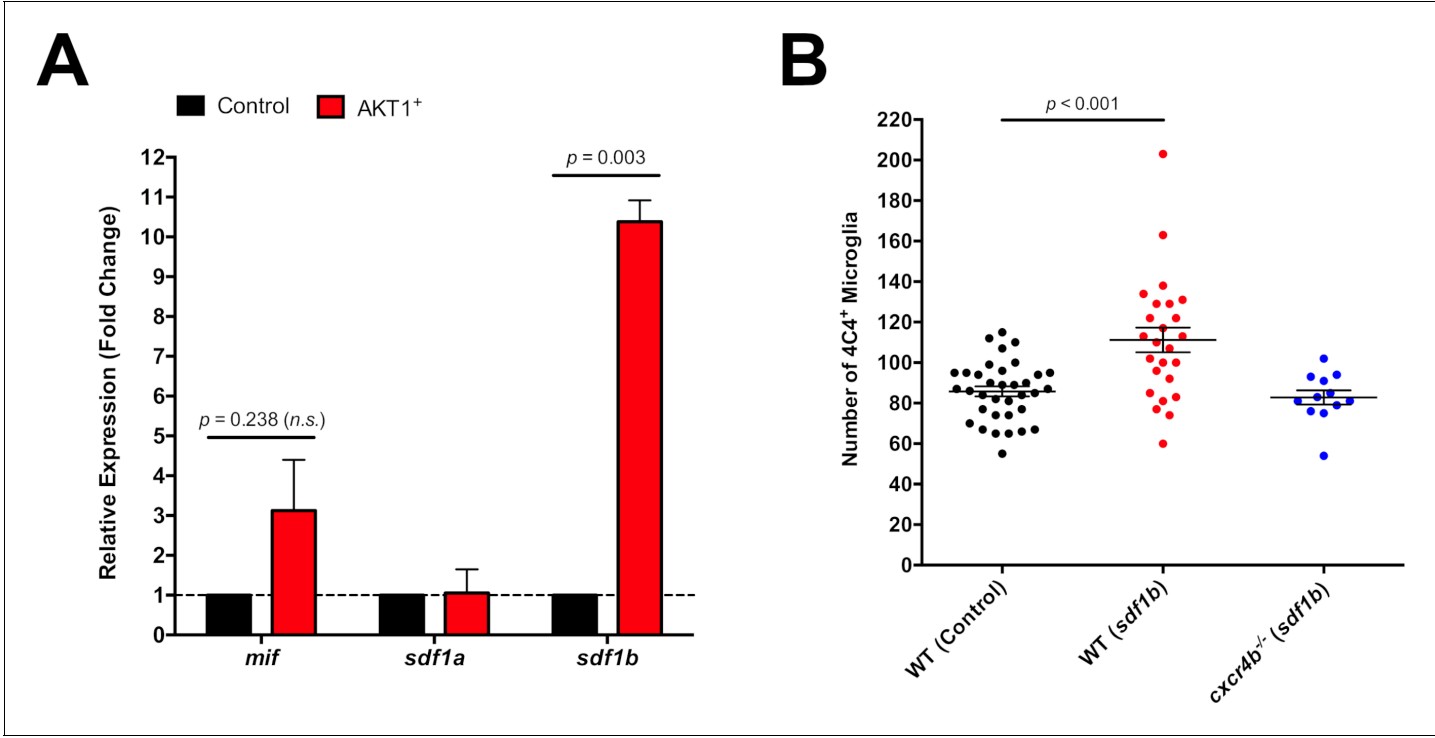

**Figure 7.** High levels of Sdf1b produced by AKT1-expressing cells lead to increased microglia numbers. (**A**) mRNA expression levels of *mif, sdf1a*, and *sdf1b* in AKT1-RFP$^+$ cells determined by qPCR (N = 3 for each gene). Fold change was measured in relation to control-RFP$^+$ cells using the comparative (ΔΔCT) method. (**B**) Quantification of the number of microglia in control WT larvae, and following Sdf1b overexpression in both WT fish (WT Control: 85.8 ± 2.45, n = 35 larvae; WT Sdf1b: 111.0 ± 6.08, n = 25 larvae, p<0.001, N = 3) and *cxcr4b$^{-/-}$* mutants (*cxcr4b$^{-/-}$* Sdf1b: 82.8 ± 3.51, n = 12, p=0.225 (n. s.), N = 3). Error bars represent mean ±SEM.

DOI: https://doi.org/10.7554/eLife.31918.017

macrophages and microglia independently of Cxcr4b using the immunosuppressant Dexamethasone, the specific CSF-1R inhibitor Ki20227 and the zebrafish *irf8$^{-/-}$* mutant. Importantly, we observed a significant decrease in the number of proliferating AKT1 cells upon macrophage and microglia depletion. Thus, macrophages and microglia promote proliferation of pre-neoplastic cells in the brain. Microglia have been shown to release stress-inducible protein 1 (STI1) for example, which increased proliferation of glioma cells in vitro and in vivo (*Carvalho da Fonseca et al., 2014*). Furthermore, co-culture studies showed that release of TGF-β by microglia-stimulated glioma growth (*Wesolowska et al., 2008*). Whether the same signals are released by microglia to promote proliferation during early stages of transformation needs to be addressed in future studies. Importantly, the fact that microglia promote the growth of pre-neoplastic cells rather than inhibiting their growth is further supported by the live imaging experiments conducted in this study. As macrophages and microglia are professional phagocytes, they are able to phagocytose cells that are detrimental to the body. However, within the time series acquired to monitor the behavior of macrophages/microglia toward AKT1-positive cells, we did not observe phagocytosis events. Macrophages/microglia infiltrated areas with high numbers of oncogenic cells and showed continuous interactions with these cells. Interestingly, similar observations were made in rodent and zebrafish live imaging models showing direct cellular interactions between microglia and glioma cells (*Resende, 2016*; *Hamilton et al., 2016*; *Bayerl et al., 2016*; *Ricard et al., 2016*). Our study highlights that signals stimulating these cellular interactions are produced early upon activation of an oncogene and seem to persist to later stages of tumor growth. However, neither the identity of these signals nor the impact of these interactions on tumor cells is known yet. As microglia have been shown to directly interact with highly active neurons with increased Ca$^{2+}$ levels (*Li et al., 2012*), it is tempting to speculate that Ca$^{2+}$ levels might be increased in AKT1-positive cells resulting in ATP/ADP release to promote the observed interactions with microglia. Identification of the signals

mediating these interactions is of high priority now to address the functional significance of these cellular contacts.

In summary, we show for the first time that macrophage and microglia are activated during the earliest stages of oncogene activation in the brain and develop their pro-tumoural activity immediately by promoting proliferation of pre-neoplastic cells. It is tempting to speculate that similar interactions occur during tumor recurrence, thus inhibiting the underlying signals might offer additional therapeutic options.

# Materials and methods

**Key resources table**

| Reagent type (species) or resource | Designation | Source or reference | Identifiers | Additional information |
|---|---|---|---|---|
| Antibody | Anti-4C4 (mouse monoclonal) | Becker Laboratory, University of Edinburgh | | (1:50) |
| Antibody | Anti-AKT1 (rabbit polyclonal) | Fisher Scientific | Fisher Scientific: PA5-29169, RRID:AB_557535 | (1:100) |
| Antibody | Anti-Cxcr4b (rabbit polyclonal) | Abcam | abcam: ab111053, RRID:AB_10860813 | (1:100) |
| Antibody | Anti-GFAP (rabbit polyclonal) | Becker Laboratory, University of Edinburgh | | (1:500) |
| Antibody | Anti-L-plastin (rabbit polyclonal) | Feng Laboratory, University of Edinburgh | | (1:500) |
| Antibody | Anti-Mfap4 (rabbit polyclonal) | Becker Laboratory, University of Edinburgh | | (1:500) |
| Antibody | Anti-PCNA (rabbit polyclonal) | Abcam | abcam: ab18197, RRID:AB_2160346 | (1:300) |
| Antibody | Anti-Sox2 | Abcam | abcam: ab97959, RRID:AB_2341193 | (1:200) |
| Antibody | Anti-Synaptophysin (rabbit polyclonal) | Abcam | abcam: ab32594, RRID:AB_765072 | (1:100) |
| Antibody | Alexa 488- or 647 secondaries | Life Technologies | Life Technologies: A11001 (RRID:AB_138404), A21235 (RRID:AB_141693), A11008 (RRID:AB_143165), A21244 (RRID:AB_141663) | (1:200) |
| Chemical compound, drug | Dexamethasone (DEX) | Sigma-Aldrich | Sigma-Aldrich: D1756 | 100 µM, 1% DMSO |
| Chemical compound, drug | Ki20227 | Tocris | Tocris: 4481 | 25 µM, 1% DMSO |
| commercial assay or kit | Quant-iT RiboGreen RNA Assay Kit | Invitrogen | Invitrogen: R11490 | |
| Commercial assay or kit | RNeasy Plus Micro Kit | QIAGEN | QIAGEN: 74034 | |
| Commercial assay or kit | SsoAdvanced Universal SYBR Green Supermix | Bio-Rad | Bio-Rad: 1725271 | |
| Commercial assay or kit | SuperScript III First-Strand Synthesis System | Invitrogen | Invitrogen: 18080–051 | |
| Gene (*Homo sapiens*) | AKT1 | NA | ENSG00000142208 | |
| Gene (*Danio rerio*) | cxcr4b | NA | ZDB-GENE-010614–1 | |
| Gene (*D. rerio*) | sdf1b (cxcl12b) | NA | ZDB-GENE-030721–1 | |

*Continued on next page*

*Continued*

| Reagent type (species) or resource | Designation | Source or reference | Identifiers | Additional information |
|---|---|---|---|---|
| Gene (*D. rerio*) | mpeg1 | NA | ZDB-GENE-030131–7347 | |
| Recombinant DNA reagent | pDEST (Gateway vector) | Invitrogen | | |
| Recombinant DNA reagent | lexOP-AKT1-RFP (plasmid) | this paper | lexOP:AKT1-lexOP:tagRFP | Gateway vector: pDEST |
| Recombinant DNA reagent | lexOP-AKT1-BFP (plasmid) | this paper | lexOP:AKT1-lexOP:tagBFP | Gateway vector: pDEST |
| Recombinant DNA reagent | lexOP-cxcr4b (plasmid) | this paper | lexOP:cxcr4b-pA | Gateway vector: pDEST |
| Recombinant DNA reagent | lexOP-sdf1b (plasmid) | this paper | lexOP:sdf1b-lexOP:tagRFP | Gateway vector: pDEST |
| Recombinant DNA reagent | lexOP-tagRFP (plasmid) | this paper | lexOP:tagRFP-pA | Gateway vector: pDEST |
| Recombinant DNA reagent | lexOP-tagBFP (plasmid) | this paper | lexOP:tagBFP-pA | Gateway vector: pDEST |
| Recombinant DNA reagent | mpeg1-cxcr4b (plasmid) | this paper | Mpeg1:cxcr4b-pA | Gateway vector: pDEST |
| Strain, strain background (*D. rerio*) | cxcr4b$^{-/-}$ | PMID: 16678780 | cxcr4b$^{t26035/t26035}$, RRID:ZFIN_ZDB-GENO-100427-6 | |
| Strain, strain background (*D. rerio*) | irf8$^{-/-}$ | PMID: 25615614 | irf8$^{st95/st95}$, RRID:ZFIN_ZDB-GENO-150504-12 | |
| Strain, strain background (*D. rerio*) | mpeg1:EGFP | PMID: 21084707 | Tg(mpeg1:EGFP)gl22, RRID:ZIRC_ZL9940 | |
| Strain, strain background (*D. rerio*) | mpeg1:mCherry | PMID: 21084707 | Tg(mpeg1:mCherry)gl23, RRID:ZIRC_ZL9939 | |
| Strain, strain background (*D. rerio*) | NBT:ΔlexPR-lexOP-pA | this paper | Tg(Xla.Tubb:LEXPR)Ed7, ZDB-ALT-180108–4 | |
| Strain, strain background (*D. rerio*) | p2ry12:p2ry12-GFP | PMID: 22632801 | TgBAC(p2ry12:p2ry12-GFP)hdb3, RRID:ZFIN_ZDB-ALT-121109-2 | |
| Software, algorithm | Imaris 8.0.2 | Bitplane | RRID:SCR_007370 | |
| Software, algorithm | LightCycler 96 Software | Roche | RRID:SCR_012155 | |

## Zebrafish maintenance

Zebrafish were housed in a purpose-built zebrafish facility, in the Queen's Medical Research Institute, maintained by the University of Edinburgh Biological Resources. All zebrafish larvae were kept at 28°C on a 14 hr light/10 hr dark photoperiod. Embryos were obtained by natural spawning from adult Tg(mpeg1:EGFP)gl22 referred to as mpeg1:EGFP, Tg(mpeg1:mCherry)gl23 referred to as mpeg1:mCherry (*Ellett et al., 2011*) wild type (AB), irf8$^{st95/st95}$ referred to as irf8$^{-/-}$ (*Shiau et al., 2015*), TgBAC(p2ry12:p2ry12-GFP)hdb3 referred to as p2ry12:p2ry12-GFP (*Sieger et al., 2012*) and cxcr4$^{t26035/t26035}$ referred to as cxcr4b$^{-/-}$ (*Haas and Gilmour, 2006*) zebrafish strains. Tg(Xla.Tubb: LEXPR)Ed7 referred to as NBT:ΔlexPR-lexOP-pA (NBT:ΔlexPR) transgenic fish were newly generated using Tol2-mediated transgenesis as described before (*Kwan et al., 2007*). ΔlexPR (Mid Entry vector; Gateway Invitrogen), a constitutively active form of the inducible LexPR system (*Emelyanov and Parinov, 2008*; *Mazaheri et al., 2014*) was placed under control of the neural-specific beta tubulin (NBT promoter (5' Entry vector; Gateway Invitrogen). Embryos were raised at 28.5°C in embryo medium (E3) and treated with 200 μM 1-phenyl 2-thiourea (PTU) (Sigma) from the end of the first day of development for the duration of the experiment to inhibit pigmentation. Animal experimentation was approved by the ethical review committee of the University of Edinburgh and the Home Office, in accordance with the Scientific Procedure Act 1986.

## DNA injections to induce oncogene expression and cellular transformation

To achieve transient expression of AKT1, zebrafish embryos were injected at the 1 cell stage. Approximately 2 nL of plasmid DNA (30 ng/µL) containing Tol2 capped mRNA (20 ng/µL) and 0.2% phenol red were injected into the eggs of NBT:ΔlexPR-lexOP-pA fish. To obtain AKT1 expression, Tol2-pDEST-lexOP:AKT1-lexOP:tagRFP or Tol2-pDEST-lexOP:AKT1-lexOP:tagBFP was injected. To obtain control RFP or BFP expression Tol2-pDEST-lexOP:tagRFP-pA or Tol2-pDEST-lexOP:tagBFP-pA were injected respectively. In embryos absent of the transgenic NBT promoter, a Tol2-pDEST-NBT:ΔlexPR-lexOP-pA (20 ng/µL) plasmid was co-injected to drive AKT1 expression in neural cells. Larvae were screened at 2 days post-fertilization (dpf) for positive transgene expression and selected for the required experiments.

To rescue Cxcr4b expression in macrophages or neural cells in the *cxcr4b*$^{-/-}$ mutant fish, a cell specific rescue construct was injected in addition to the NBT driver and lexOP:tagRFP-pA/lexOP:*AKT1*-lexOP:tagRFP constructs. Cxcr4b expression was recovered in macrophages through the mpeg1 promoter ('Macrophage' rescue) via a mpeg1:*cxcr4b*-pA construct. The expression of cxcr4b in neural cells ('Neural' rescue) was rescued through the lexOP:*cxcr4b*-pA construct.

## Whole mount immunohistochemistry, in situ hybridization, image acquisition and live imaging

Whole mount immunostaining of samples was performed as previously described (*Astell and Sieger, 2017*). Briefly, larvae were fixed in 4% PFA/1% DMSO at room temperature for 2 hr, followed by a number of washes in PBStx (0.2% Triton X-100 in 0.01 M PBS), and blocked in 1% goat serum blocking buffer (1% normal goat serum, 1% DMSO, 1% BSA, 0.7% Triton X-100 in 0.01 M PBS) for 2 hr prior to incubation with primary antibodies overnight at 4˚C. Primary antibodies used were mouse anti-4C4 (1:50) (courtesy of Becker Laboratory, University of Edinburgh), rabbit anti-AKT1 (1:100) (PA5-29169, Fisher Scientific), rabbit anti-Cxcr4b (1:100) (ab111053, abcam), rabbit anti-GFAP (1:500) (courtesy of Becker Laboratory, University of Edinburgh), rabbit anti-L-plastin (1:500) (courtesy of Feng Laboratory, University of Edinburgh), rabbit anti-Mfap4 (1:500) (courtesy of Becker Laboratory, University of Edinburgh),rabbit anti-PCNA (1:300) (ab18197, abcam), rabbit anti-SOX2 (1:200) (ab97959, abcam), and rabbit anti-Synaptophysin (1:100) (ab32594, abcam). A number of washes in PBStx was carried out before samples were subsequently incubated in conjugated secondary antibodies (goat anti-mouse Alexa Fluor 488 [1:200]; goat anti-mouse Alexa Fluor 647 [1:200]; goat anti-rabbit Alexa Fluor 488 [1:200]; goat anti-rabbit Alexa Fluor 647 [1:200]) (Life Technologies) overnight at 4˚C to reveal primary antibody localizations. Samples were washed following secondary antibody incubation and kept in 70% glycerol at 4˚C until final mounting in 1.5% low melting point agarose (Life Technologies) in E3 for image acquisition.

Whole brain immuno-fluorescent images were acquired using confocal laser scanning microscopy (Zeiss LSM710 and LSM780; 20x/0.8 objective; 2.30 µm intervals; 488-, 543-, and 633 nm laser lines).

Whole mount in situ hybridization was done following the protocol described by Thisse and Thisse (*Thisse and Thisse, 2008*) using a digoxygenin labeled *mpeg1* RNA probe.

Live imaging of zebrafish larvae was performed as previously described (*Hamilton et al., 2016*; *Sieger et al., 2012*); samples were mounted dorsal side down in 1.5% low melting point agarose (Life Technologies), in glass-bottom dishes (MatTek) filled with E3 containing 0.2 mg/mL Tricaine (MS222, Sigma). Single time-point live images were acquired through confocal imaging (Zeiss LSM710; 20x/0.8 objective; 2.30 µm intervals; 405-, 488-, and 543 nm laser lines). To investigate direct interactions between oncogene-expressing cells and microglia, time-lapse imaging was carried out on a spinning disk confocal microscope (Andor iQ3; 20x/0.75 and W40x/1.15 objectives; 1.5–2 µm z-intervals; 405-, 488-, and 543 nm laser lines). All time-lapse acquisitions were carried out in temperature-controlled climate chambers set to 28˚C for 10–18 hr.

## Histology and immunofluorescence

Twenty zebrafish, injected as previously described to obtain transient expression of AKT1 in neural cells, were used at 1 month of age to assess the presence of proliferative lesions. The brains were observed and photographed immediately upon removal under a fluorescent stereomicroscope and fixed in 4% PFA in PBS for 12 hr. After sucrose treatment and OCT embedding, 12 um thick sections

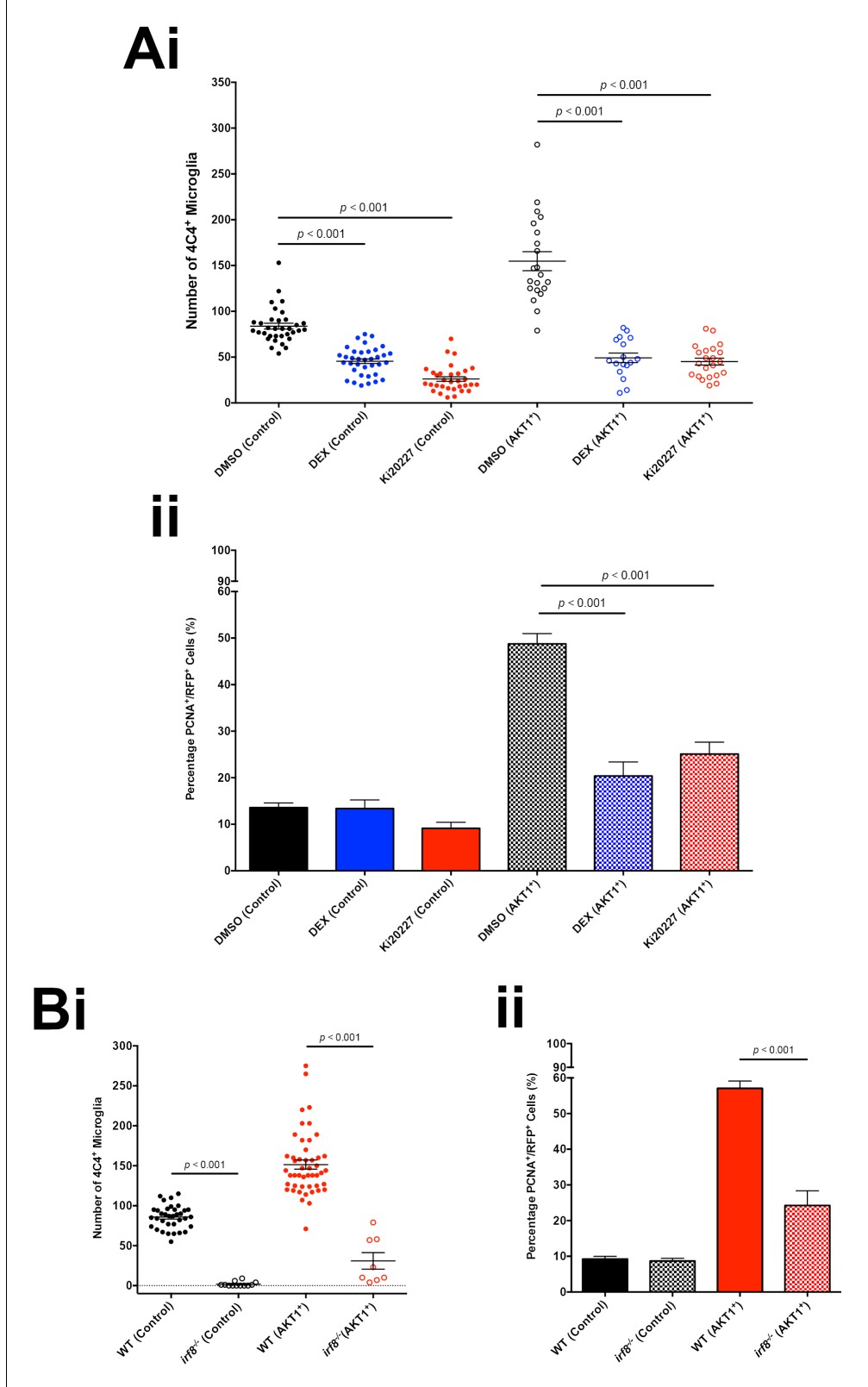

**Figure 8.** Microglia promote proliferation of AKT1-expressing cells. (A) Treatment with the immunosuppressant Dexamethasone (DEX) and the CSF-1R inhibitor Ki20227 led to reduced microglia numbers and proliferation rates of AKT1-expressing cells in the zebrafish larvae. (i) Quantification of the number of microglia in control and upon AKT1 overexpression in DMSO controls and fish treated with DEX (DMSO Control: 83.8 ± 3.29, n = 34 larvae; DEX Control: 45.5 ± 2.53, n = 36 larvae, p<0.001, N = 3; DMSO AKT1: 154.7 ± 10.38, n = 21 larvae; DEX AKT1: 49.2 ± 5.16, n = 17 larvae, p<0.001,

*Figure 8 continued on next page*

*Figure 8 continued*

N = 3) and fish treated and Ki20227 (DMSO Control: 83.8 ± 3.29, n = 34 larvae; Ki20227 Control: 26.1 ± 2.55, n = 32 larvae, p<0.001, N = 3; DMSO AKT1: 154.7 ± 10.38, n = 21 larvae; Ki20227 AKT1: 45.1 ± 3.75, n = 22 larvae, p<0.001, N = 3) (ii) Quantification of the level of proliferation of RFP-expressing cells in control and AKT1 overexpression in DMSO controls and fish treated with DEX (DMSO Control: 13.6 ± 1.01%, n = 20; DEX Control: 13.4 ± 1.85%, n = 15 larvae, p=1.00 (n.s.), N = 3; DMSO AKT1: 48.8 ± 2.23%, n = 20 larvae; DEX AKT1: 20.4 ± 3.02%, n = 11 larvae, p<0.001, N = 3) and fish treated with Ki20227 (DMSO Control: 13.6 ± 1.01%, n = 20 larvae; Ki20227 Control: 9.14 ± 1.30%, n = 17 larvae, p=1.00 (n.s.), N = 3; DMSO AKT1: 48.8 ± 2.23%, n = 20 larvae; Ki20227 AKT1: 25.1 ± 2.54%, n = 20 larvae, p<0.001, N = 3) Error bars represent mean ±SEM. (B) *irf8*[-/-] zebrafish larvae showed reduced microglia numbers and proliferation rates of AKT1-expressing cells. (i) Quantification of the number of microglia in control and AKT1-positive brains in wt and *irf8*[-/-] mutant zebrafish larvae (Control WT: 85.8 ± 2.45, n = 35 larvae; Control *irf8*[-/-]: 6.25 ± 2.76, n = 20 larvae, N = 3, p<0.001) (AKT1 WT: 151.4 ± 5.70, n = 48 larvae; AKT1 *irf8*[-/-]: 31.0 ± 10.32, n = 8 larvae, p<0.001, N = 3). (ii) Quantification of the level of proliferation of RFP-expressing cells in control and AKT1-positive brains in wt and *irf8*[-/-] mutant zebrafish larvae (Control WT: 9.25 ± 0.75%, n = 13; Control *irf8*[-/-]: 8.69 ± 0.76%, n = 12, p<1.00 (n.s.), N = 3) (AKT1 WT: 57.1 ± 2.03%, n = 17; AKT1 *irf8*[-/-]: 24.3 ± 4.11%, n = 8, p<0.001, N = 3).

DOI: https://doi.org/10.7554/eLife.31918.018

were cut and collected, to allow staining of adjacent sections for different antibodies. Immunostaining was performed as previously described (*Mayrhofer et al., 2017*). Sections were counterstained with DAPI and observed through confocal imaging (Zeiss LSM710; 40x/1.3 objective).

## Image analysis and quantifications

Analyses of all images were conducted using Imaris (Bitplane, Zurich, Switzerland). For the quantification of 4C4[+] and/or L-plastin[+] cells, only cells within the brain (telencephalon, tectum, and cerebellum) were counted for each sample using the 'Spots' function tool in Imaris 8.0.2. To determine 4C4[+]/L-plastin[+] cells, the 'Coloc' function in Imaris was used to generate an independent channel for cells that were double-positive. To quantify proliferation rates, the number of PCNA[+]/RFP[+] cells were counted in relation to the total number of RFP[+] cells and the averaged value expressed as measure of percentage proliferation. To assess and quantify microglial morphology, we used the surface rendering tool in Imaris 8.0.2. This allowed segmentation of individual cells in 3D. To assess their morphological activation we calculated the ratio of the cellular surface and cellular volume of individual cells as previously described (*Gyoneva et al., 2014*). Microglia with a ratio smaller than 0.8 were classified as activated.

## Cell isolation, FACS and Q-PCR

For cell isolation, anaesthetized 8 dpf zebrafish larvae were transferred into ice cold E3 containing 0.2 mg/mL Tricaine. The heads were removed using surgical micro-scissors and then collected in ice cold medium A (HBSS 1X, 15 mM Hepes and 25 mM Glucose (Life technologies)). On ice, heads were manually homogenized using a 1 mL glass dounce until the cell suspension was homogenous. Subsequently, the cell suspension was run through a 40 μm cell strainer and transferred into a 1.5-mL Eppendorf tube. Cells were centrifuged at 300 g for 10 min at 4℃. The pellet was re-suspended in 22% percoll solution overlaid by ice cold PBS, followed by centrifugation at 950 g without brake for 30 min at 4℃. The pellet was washed twice with medium A supplemented with 2% normal goat serum and centrifuged at 300 g for 10 min at 4℃. Cells were re-suspended in ice cold medium A - 2% normal goat serum then transferred to FACS tubes through 35 μm cell strainer cap, immediately followed by FACS sorting using a FACSAria II (Becton Dickinson). DAPI was added to label and exclude dead cells.

Total RNA extraction from the sorted cells of interest was performed using the Qiagen RNeasy Plus Micro kit (Qiagen) and the RNA sample concentration was determined using the Quant-IT Ribogreen RNA assay kit (Invitrogen).

For quantification of gene expression, cDNA synthesis following RNA isolation was carried out using the SuperScript® III First-Strand Synthesis System (Invitrogen). Quantitative PCR (Q-PCR) amplifications were performed in duplicates in a 20 μL reaction volume containing SsoAdvanced Universal SYBR Green Supermix (Bio-Rad) using a LightCycler 96 Real-Time PCR System (Roche). The PCR protocol used was: Initial denaturation step of 5 min at 95℃, and 45 cycles of 10 s at 95℃, 20 s at 56℃, and 20 s at 72℃. Primers used were: *ß-actin* forward: 5'-CACTGAGGCTCCCCTGAATCCC-3', *ß-actin* reverse: 5'-CGTACAGAGAGA GCACAGCCTGG-3'; *mif* forward: 5'-AAAGACTCGG TTCCGGCG-3', *mif* reverse: 5'-CACACGGGTCTCCTTTTCCC-3'; *sdf1a* forward: 5'-CGCCAT

TCATGCACCGATTT-3', *sdf1a* reverse: 5'-TGACTTGGAAGGGGCAGTTG-3', *sdf1b* forward: 5'-AGCAAAGTAGTAGCGCTGGTG-3', and *sdf1b* reverse: 5'-TCTCTCGGATGCTCCGTTG-3'. Melting curve analysis was used to ensure primer specificity. For Q-PCR analysis, the threshold cycle (Ct) values for each gene were normalized to expression levels of ß-actin and relative quantification of gene expression determined with the comparative Ct (ΔΔCt) method using the LightCycler® 96 Software (Roche). Q-PCR analysis was repeated three times for each gene.

## Pharmacological treatments

To suppress the immune response, larvae were treated with dexamethasone (DEX, Sigma-Aldrich). Briefly, larvae were incubated in 100 µM DEX/1% DMSO from 3 dpf until the end of the experiment. To deplete macrophages and microglia, larvae were treated with the CSF-1R inhibitor Ki20227 (Tocris) at 25 µM/1% DMSO from 3 dpf until the end of the experiment. To obtain experimental controls, age-matched samples were incubated in 1% DMSO. Media were changed daily until the respective experimental endpoints at 8 dpf.

## Statistical analysis

All experiments were performed in at least two replicates and 'n' indicates the total number of larvae used. All measured data were analyzed (StatPlus, AnalystSoft Inc.). Two-tailed Student's *t*-tests were performed between two experimental groups and one-way ANOVA with Bonferroni's post-hoc tests performed for comparisons between multiple experimental groups. Statistical values of $p < 0.05$ were considered to be significant. All graphs were plotted in Prism 6.1 (GraphPad Software) and values presented as population means (±SEM).

## Acknowledgements

The authors thank the BRR zebrafish facility (QMRI, University of Edinburgh) for maintenance and care of the zebrafish. The authors are grateful to members of the CALM and SURF facilities (University of Edinburgh) for assistance with microscope imaging and FACS. The authors thank Francesca Peri for providing access to laboratory space and equipment. The authors are grateful to Graham Lieschke for sharing mpeg1:EGFP fish and to Darren Gilmour for sharing *cxcr4b*−/− zebrafish. pcDNA3 Myr HA Akt1 was a gift from William Sellers (Addgene plasmid # 9008). Thanks to Yi Feng for providing the L-plastin antibody and to Catherina and Thomas Becker for sharing the 4C4 antibody. We thank Katy Astell, David Lyons and Liz Patton for critical reading of this article. D.S. was supported by a Cancer Research UK Career Establishment Award.

## Additional information

### Funding

| Funder | Grant reference number | Author |
|---|---|---|
| Cancer Research UK | Career Establishment Award, C49916/A17494 | Dirk Sieger |

The funders had no role in study design, data collection and interpretation, or the decision to submit the work for publication.

### Author contributions

Kelda Chia, Marina Mione, Investigation, Visualization, Methodology, Writing—review and editing; Julie Mazzolini, Investigation, Methodology, Writing—review and editing; Dirk Sieger, Conceptualization, Resources, Supervision, Funding acquisition, Investigation, Methodology, Writing—original draft, Writing—review and editing

### Author ORCIDs

Marina Mione (ID) http://orcid.org/0000-0002-9040-3705
Dirk Sieger (ID) http://orcid.org/0000-0001-6881-5183

## Ethics

Animal experimentation: Animal experimentation was reviewed and approved by the ethical review committee of the University of Edinburgh and the Home Office (Project license 60/4544), in accordance with the Scientific Procedure Act 1986.

## Decision letter and Author response

Decision letter https://doi.org/10.7554/eLife.31918.021
Author response https://doi.org/10.7554/eLife.31918.022

## Additional files

### Supplementary files

• Transparent reporting form
DOI: https://doi.org/10.7554/eLife.31918.019

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
