## [Decision Letter]

Thank you for submitting your article "Brain tumor initiating cells induce Cxcr4 mediated infiltration of pro-tumoral macrophages into the brain" for consideration by *eLife*. Your article has been favorably evaluated by Tadatsugu Taniguchi (Senior Editor) and three reviewers, one of whom is a member of our Board of Reviewing Editors. The following individuals involved in review of your submission have agreed to reveal their identity: Jean-Pierre Levraud (Reviewer #2); Charles K Kaufman (Reviewer #3).

The reviewers have discussed the reviews with one another and the Reviewing Editor has drafted this decision to help you prepare a revised submission.

Summary:

In this manuscript, Chia et al. demonstrate that AKT activation in neuronal populations leads to an increase in microglial numbers in the developing brain. They argue that this is due to an infiltration of primitive macrophages into the brain after AKT activation, and that this is mediated by an sdf1b/cxcr4b axis. They extend this argument to suggest that these microglial cells are pro-tumorigenic, as has been previously demonstrated in murine models of glioma. The data to indicate that AKT1 activation leads to an early increase in microglial cells is the most convincing. There are several issues that could improve this manuscript:

Major comments:

1) The early effects of NBT-AKT1

In Figure 1 and Figure 1—figure supplement 1, the authors argue that AKT1 activation gives rise to a mixed neuronal/glial pre-neoplastic tumor. Yet the promoter they used is putatively a neuronal driver, making it unclear to me why the glial population is increased as well. Where are these cells arising from? Along these lines, there is no quantification of the neuronal vs. glial populations to know how important each of these populations are. Are these glial populations important for the microglial effects as well? Finally, the panels in Figure 1—figure supplement 1 are very difficult to understand, and need to be labeled much more explicitly as to what each panel represents (i.e. what is L and what is M, etc.).

2) The effects of AKT1 on microglial activation

In Figure 2, it is very difficult to discern what is morphologically different in the AKT1 overexpressing cells compared to controls. The quantification seems to indicate activation phenotypes, but to a non-expert in this morphology this would be difficult to understand. The method for morphology-based assessment of the activated state of microglia is not specified and needs to be quantified explicitly. A much higher resolution image of one or 2 cells, with clear delineation of what was scored in panel 2C would be more useful.

Along the same lines, in Figure 3 and Video 1 and Video 2 the images seem consistent with the idea that the microglial/neuronal cells stay in contact with each other, but it seems just as possible that there are simply more microglial cells in general, thus less room for them to move around. Are the increased interactions between microglia/macrophages and Akt-1 expressing neural cells due to increased numbers/size of the Akt-1 expressing cells (just more chance of "bumping" into each other) or is there truly a functional relationship with Cxcr4-Sdf1b interactions? How do the authors quantify these effects of persistence? What do they propose is the importance of such persistence?

3) The recruitment of primitive macrophages into the brain

The data in Figure 4 is somewhat confusing and not that convincing for recruitment versus differentiation. First, because of the curvature of the zebrafish brain, some areas of the maximally projected picture must include zones outside of the brain parenchyma. The area that correspond to parenchyma only should be delimited. Next, they propose that the first cells are recruited L-plastin^+^/4C4^-^ primitive macrophages, which then eventually differentiate into L-plastin^+^/4C4^+^ microglia. But this type of analysis would require a time course showing that these newly arrived cells come first, and then differentiate into the 4C4^+^ population, and I cannot see what type of time course was done. It seems like all the staining was done at 8dpf, so how can one argue that the cells are first recruited from the periphery and then differentiate in situ. Is there a transgenic marker for the peripheral primitive macrophages which would allow for time-lapse imaging to show that these cells are indeed recruited into the brain after AKT1 activation?

Finally, an alternative explanation for the reduced infiltration of macrophages in the *cxcr4b* mutant could be that these fish possess fewer peripheral macrophages than siblings. Can you assess their population to exclude this possibility?

4) The role of microglial cells in proliferation of AKT1 positive cells

In Figure 5 and Figure 6, the authors indicate that CXCR4b might be necessary for proliferation of the AKT1^+^ cells (Figure 5). To then delineate whether this was through the microglia or not, they perform macrophage versus neuronal rescue experiments, which is a nice way of getting at the question. Confusingly, however, the authors state "…the specific rescue of Cxcr4b in macrophages and microglia led to increased albeit not significantly, proliferation rates of AKT1 positive cells". This statement seems to indicate that the authors feel that the effect of Cxcr4b on the AKT1 positive cells is mediated by macrophages, but don't provide evidence this is the case and in fact do not report the control versus macrophage rescue data (only the macrophage rescue at 35.3+-2.83%) with a p=1. Is this because the rescue was mosaic so you don't expect rescue of this effect?

Similarly, Figure 8 to shows that the macrophages/microglia are required for AKT1^+^ tumor proliferation, despite the fact that the CXCR4b data seems to argue against this. In these experiments, the authors state that AMD3100 showed reduced proliferation of AKT1 cells (from 48 to 24%, i.e. a 50% reduction). They then state that control cells also showed a "slightly reduced proliferation", but the numbers they report go from 13.6% to 6.8%, i.e. also about a 50% decrease. In essence the effect of AMD3100 is not really different in the control versus AKT1 cells. Likewise, both the dexamethasone and Ki20227 data alone do not really demonstrate whether this reduction of AKT1^+^ proliferation is via the microglia/macrophages either, since it could be a more direct effect on the tumor cells themselves. A clearer explanation of these effects on proliferation would be helpful.

5) The Akt-Sdf1 axis

Akt sits at the nexus of many signaling pathways – can the authors speculate on how its activation modulates Sdf1 expression in the neural cell?

---

## [Author Response]

Major comments:1) The early effects of NBT-AKT1In Figure 1 and Figure 1—figure supplement 1, the authors argue that AKT1 activation gives rise to a mixed neuronal/glial pre-neoplastic tumor. Yet the promoter they used is putatively a neuronal driver, making it unclear to me why the glial population is increased as well. Where are these cells arising from? Along these lines, there is no quantification of the neuronal vs. glial populations to know how important each of these populations are. Are these glial populations important for the microglial effects as well? Finally, the panels in Figure 1—figure supplement 1 are very difficult to understand, and need to be labeled much more explicitly as to what each panel represents (i.e. what is L and what is M, etc.).

Indeed, the immunohistochemistry on tumors at 30 dpf revealed brain tumors with mixed neural and glial components. However, the glial component (tested via GFAP immunostaining) develops only at later time points and is not present at the early time points when the microglia response is detected. We provide new data in the revised version of the manuscript showing that at 8 dpf the pre-neoplastic cells are negative for GFAP (Figure 1). As we detect a prominent microglial response at this time point, we conclude that the glial populations are not required for the microglial effects observed. Furthermore, we provide new data showing that at 8 dpf a subset of the pre-neoplastic AKT1 cells were positive for the stem cell marker Sox2, while none of the neural control cells appeared to be positive for Sox2 (Figure 1). This is suggestive of cellular dedifferentiation caused by oncogene activation. This is a phenomenon that was described a few years ago in a rodent glioma model (Friedmann-Morvisnki et al., 2012). Friedmann-Morvisnki and colleagues showed that oncogenic alterations can induce dedifferentiation of mature neurons and astrocytes. These dedifferentiated cells generate a NSC or progenitor state which then gives rise to the heterogeneous populations in brain tumors at later stages.

Thus, we assume that we observed a similar phenomenon in our model where AKT1 induces dedifferentiation of neural cells (reflected in Sox2 activation) and these cells give rise to the heterogeneous cell population within tumors at 30 dpf. We have added this explanation to the Results now (subsection “Expression of human AKT1 induces cellular transformation in the larval zebrafish brain”).

We have rearranged Figure 1—figure supplement 1 and provide improved labeling to allow an easier understanding of the figure.

2) The effects of AKT1 on microglial activationIn Figure 2, it is very difficult to discern what is morphologically different in the AKT1 overexpressing cells compared to controls. The quantification seems to indicate activation phenotypes, but to a non-expert in this morphology this would be difficult to understand. The method for morphology-based assessment of the activated state of microglia is not specified and needs to be quantified explicitly. A much higher resolution image of one or 2 cells, with clear delineation of what was scored in panel 2C would be more useful.

We apologize for not providing a detailed description of the morphology-based assessment of the microglial activation state in the manuscript. To assess and quantify microglial morphology we used the surface rendering tool in Imaris (Bitplane). This allows segmentation of individual cells in 3D. Based on this we calculated the ratio of the cellular surface and cellular volume of individual cells. Ramified (non-activated) microglia show a relatively small cellular volume and thus a surface/volume ratio of ~1. In contrast, amoeboid microglia show a surface/volume ratio of ~0.6. As there is a continuum of transition steps from ramified to amoeboid, we set a threshold and considered all cells with a ratio smaller 0.8 as activated. This approach has been used and published by others in the past (Gyoneva et al., 2014). In the revised version, we provide higher resolution images of the different morphologies of microglia (Figure 2). In addition, we provide segmented images generated with Imaris (Bitplane) showing the surface application on representative cells. We have also included a description of the quantifications in the Materials and methods and provide a reference to the previously published work from Gyoneva et al., using this method (subsection “Image analysis and quantifications”).

Along the same lines, in Figure 3 and Video 1 and Video 2 the images seem consistent with the idea that the microglial/neuronal cells stay in contact with each other, but it seems just as possible that there are simply more microglial cells in general, thus less room for them to move around. Are the increased interactions between microglia/macrophages and Akt-1 expressing neural cells due to increased numbers/size of the Akt-1 expressing cells (just more chance of "bumping" into each other) or is there truly a functional relationship with Cxcr4-Sdf1b interactions? How do the authors quantify these effects of persistence? What do they propose is the importance of such persistence?

This is a very interesting point and we agree that the higher density of cells could enforce interactions by chance. However, many of these direct cellular surface interactions were observed for several hours without any interruptions, thus it is unlikely that these were random contacts simply enforced by the high density of microglial and AKT1 positive cells. To address this further, we provide a new data set in the revised version. Here, we injected lower amounts of the lexOP:*AKT1*-lexOP:tagRFP construct into oocytes of double transgenic mpeg1:EGFP/NBT:ΔLexPR fish and screened for fish with lower numbers and isolated AKT1 positive cells. In these larvae microglia were frequently seen to interact with AKT1 positive cells (Figure 3; Video 3 and Video 4 = 6 samples analyzed) (subsection “Microglia show highly dynamic interactions with AKT1 positive cells”). These interactions lasted for several hours and were not seen with control cells (not shown). Thus, we conclude that microglia are pro-actively interacting with AKT1 positive cells. Importantly, direct cellular surface interactions between microglia and tumor cells were observed by others in recently published rodent glioma models as well (Resende et al. 2016; Bayerl et al. 2016; Ricard et al. 2016). Furthermore, we observed direct cellular interaction between microglia and human glioblastoma cells in a recently published xenograft model (Hamilton et al., 2016). We discuss these findings in the Discussion of the manuscript (second to last paragraph). However, the underlying mechanisms stimulating these interactions have not been addressed yet and the functional importance of these cellular contacts is not known. In the revised version of the manuscript we speculate on a putative role for Ca^2+^/ATP signaling as microglia have been shown to directly interact with highly active neurons with increased Ca^2+^ levels (Li et al., 2012). Future studies will be needed to identify the underlying mechanism and to address a putative functional impact of these interactions.

3) The recruitment of primitive macrophages into the brainThe data in Figure 4 is somewhat confusing and not that convincing for recruitment versus differentiation. First, because of the curvature of the zebrafish brain, some areas of the maximally projected picture must include zones outside of the brain parenchyma. The area that correspond to parenchyma only should be delimited. Next, they propose that the first cells are recruited L-plastin^+^/4C4^-^ primitive macrophages, which then eventually differentiate into L-plastin^+^/4C4^+^ microglia. But this type of analysis would require a time course showing that these newly arrived cells come first, and then differentiate into the 4C4^+^ population, and I cannot see what type of time course was done. It seems like all the staining was done at 8dpf, so how can one argue that the cells are first recruited from the periphery and then differentiate in situ. Is there a transgenic marker for the peripheral primitive macrophages which would allow for time-lapse imaging to show that these cells are indeed recruited into the brain after AKT1 activation?

We agree that the presentation of the maximum intensity projected images was slightly confusing. As stated in the Materials and methods (subsection “Image analysis and quantifications”), only cells within the brain parenchyma were counted. We have demarcated the outline of the brain parenchyma in the revised version of Figure 4 to illustrate this and specified this within the text (subsection “Macrophage infiltration in response to AKT1 induced transformation leads to increased numbers of 4C4 microglia”, first paragraph) and Figure 4 legend.

We are grateful for the suggestion to provide a time course to identify a time point when L-plastin^+^/4C4^-^ primitive macrophages have been recruited to the brain but have not differentiated into L-plastin^+^/4C4^+^ microglia yet. We have analyzed infiltration and differentiation at 3 dpf and added this new data set to Figure 4. Indeed, at 3 dpf we detect a significant increase of L-plastin^+^/4C4^-^ primitive macrophages in AKT1 positive brains compared to controls (Figure 4Ci). Intriguingly, we do not detect an increase in L-plastin^+^/4C4^+^ microglia at this time point (Figure 4Cii). Thus, these newly infiltrated cells have not differentiated. As we detect an increase in L-plastin^+^/4C4^+^ microglia at later time points in the absence of proliferation, these new results further support our conclusion that newly infiltrated L-plastin^+^/4C4^-^ macrophages differentiate into L-plastin^+^/4C4^+^ microglia like cells over time.

To address this point even further we performed live imaging experiments upon AKT1 overexpression in a double transgenic model and overexpressed AKT1 in p2ry12:p2ry12-GFP/mpeg1:mCherry zebrafish. In these zebrafish, all macrophages (including microglia) are positive for mCherry and microglia can be identified based on their additional P2ry12-GFP expression. Intriguingly, we detected p2ry12-GFP^-^/mCherry^+^ macrophages at the dorsal periphery of AKT1 positive brains (Figure 4—figure supplement 1) at 5.5 dpf. Eventually, some of these macrophages infiltrated the brain parenchyma and started expressing GFP over time, showing their differentiation towards a p2ry12-GFP^+^/mCherry^+^ microglia-like cell (Figure 4—figure supplement 1, arrowheads, Video 5).

In summary, these two new data sets strongly support our conclusion that peripheral macrophages are recruited to the brain and differentiate towards microglia-like cells upon AKT1 overexpression.

Finally, an alternative explanation for the reduced infiltration of macrophages in the cxcr4b mutant could be that these fish possess fewer peripheral macrophages than siblings. Can you assess their population to exclude this possibility?

Cxcr4b mutant zebrafish do neither show a general deficit in the number of peripheral macrophages nor in their function. We provide new data in the revised version of the manuscript to compare peripheral macrophages in wt and *cxcr4b* mutant zebrafish (Figure 5—figure supplement 1, subsection “Cxcr4b signaling is required for the increase in macrophage and microglial numbers”, first paragraph) and don’t detect differences. Furthermore, microglia numbers were similar in *cxcr4b^-/-^*controls and wildtype controls, showing that Cxcr4b signaling is not needed for the normal developmental population of the brain by microglia (WT Control: 85.4 ± 2.46, n = 35; *cxcr4b^-/-^*Control: 79.8 ± 2.57, n = 45, *p* = 0.103 (n.s.); Figure 5). Finally, we also refer to a recent publication showing that macrophages in *cxcr4b^-/-^*zebrafish show normal recruitment and function upon mycobacterial infections (Torraca et al., 2017). Thus, we conclude that the reduced infiltration of macrophages in the cxcr4b mutant upon AKT1 overexpression is due to the inability of the macrophages to respond to Sdf1b signaling.

4) The role of microglial cells in proliferation of AKT1 positive cellsIn Figure 5 and Figure 6, the authors indicate that CXCR4b might be necessary for proliferation of the AKT1^+^ cells (Figure 5). To then delineate whether this was through the microglia or not, they perform macrophage versus neuronal rescue experiments, which is a nice way of getting at the question. Confusingly, however, the authors state "…the specific rescue of Cxcr4b in macrophages and microglia led to increased albeit not significantly, proliferation rates of AKT1 positive cells". This statement seems to indicate that the authors feel that the effect of Cxcr4b on the AKT1 positive cells is mediated by macrophages, but don't provide evidence this is the case and in fact do not report the control versus macrophage rescue data (only the macrophage rescue at 35.3+-2.83%) with a p=1. Is this because the rescue was mosaic so you don't expect rescue of this effect?

We thank you for highlighting this point. Indeed, we concluded that Cxcr4b is specifically required in macrophages and microglia and that these cells promote proliferation of AKT1^+^ cells. As these rescue experiments are based on mosaic expression of Cxcr4b the number of samples analyzed in the previous version of the manuscript was not high enough to achieve statistically significant results. We have repeated these rescue experiments in the meantime and provide higher sample numbers for the final analysis. Importantly, the final data set confirms our conclusion and shows that the specific rescue of Cxcr4b in macrophages and microglia leads to a significant increase in the proliferation rates of AKT1 positive cells (Figure 6).

Similarly, Figure 8 to shows that the macrophages/microglia are required for AKT1^+^ tumor proliferation, despite the fact that the CXCR4b data seems to argue against this. In these experiments, the authors state that AMD3100 showed reduced proliferation of AKT1 cells (from 48 to 24%, i.e. a 50% reduction). They then state that control cells also showed a "slightly reduced proliferation", but the numbers they report go from 13.6% to 6.8%, i.e. also about a 50% decrease. In essence the effect of AMD3100 is not really different in the control versus AKT1 cells. Likewise, both the dexamethasone and Ki20227 data alone do not really demonstrate whether this reduction of AKT1^+^ proliferation is via the microglia/macrophages either, since it could be a more direct effect on the tumor cells themselves. A clearer explanation of these effects on proliferation would be helpful.

We agree that the presentation of the results in Figure 8 was not optimal and did not fully support our conclusions. We have reviewed this paragraph and figure again and have re-arranged based on the following points. Firstly, we acknowledge that AMD3100 is not an optimal tool to interfere with Cxcr4b signaling in zebrafish. Our own data in the previous version of the manuscript showed a discrepancy between AMD3100 treatment and the Cxcr4b mutant zebrafish. While the *cxcr4b* mutant zebrafish did not show a reduced proliferation rate of control cells compared to wildtype zebrafish (Figure 5), we detected a decrease in the proliferation rate of control cells upon AMD3100 treatment. Thus, we concluded that the decrease upon AMD3100 treatment must be caused by Cxcr4b independent effects of AMD3100. Furthermore, we spoke to other zebrafish researchers including experts in the Cxcr4b field, who confirmed that they were not able to phenocopy the zebrafish *cxcr4b* mutant with AMD3100. In conclusion, we realized that the use of AMD3100 did not provide any new insights compared to the *cxcr4b* mutant data (Figure 5) but rather blurred the clear data generated with the mutant. Thus, we decided to remove the AMD3100 data set from the manuscript.

Instead we performed a Cxcr4b independent depletion of macrophages and microglia to address their impact on the proliferation of AKT1^+^ cells. Treatment with Dexamethasone and Ki20227 lead to a robust depletion of macrophages and microglia (Figure 8Ai). Furthermore, both compounds significantly decreased the proliferation rate of AKT1^+^ cells (Figure 8Aii). As the treatment with Dexamethasone, Ki20227 and DMSO were done in parallel in the same experiment we present the results together in one graph now and performed a one-way ANOVA with Bonferroni’s post-hoc tests to analyze statistical significance.

Finally, to address the impact of macrophages and microglia on the proliferation of AKT1^+^ cells without any pharmacological interference we made use of the zebrafish *irf8^-/-^*mutant. The *irf8* null mutant (*irf8^-/-^*) zebrafish was characterized to lack macrophages up to around 6 dpf, with recovery from 7 dpf while microglia were absent in the brain until 31 dpf (Shiau et al., 2015). In line with published observations we did not detect 4C4 positive microglial cells in *irf8^-/-^*larvae at 8dpf (Figure 8Bi). Interestingly, upon AKT1 overexpression in *irf8^-/-^* larvae we detected a small population of 4C4 positive cells (Figure 8Bi). These cells were probably derived from the population of macrophages that recover in *irf8^-/-^* larvae from 7 dpf, which then infiltrated into the brain due to AKT1 overexpression and differentiated into 4C4^+^ cells. Importantly, the number of 4C4 positive cells was 80% lower in *irf8^-/-^*larvae compared to wildtype larvae upon AKT1 overexpression (Figure 8Bi), which allowed us to address the impact on the proliferation of AKT1 positive cells. While proliferation rates of neural cells remained constant in *irf8^-/-^* controls (Figure 8Bii), we detected an almost 60% reduction in proliferation of AKT1 positive cells compared to AKT1 positive cells in wildtype larvae (Figure 8Bii).

In summary, the combination of the pharmacological data and the *irf8^-/-^* mutant data allows us to conclude that macrophages and microglia promote proliferation of AKT1 positive cells in the larval zebrafish brain.

5) The Akt-Sdf1 axisAkt sits at the nexus of many signaling pathways – can the authors speculate on how its activation modulates Sdf1 expression in the neural cell?

We provide a speculation on how AKT1 modulates Sdf1 expression in the Discussion of the revised manuscript (second paragraph). In brief, previous studies revealed an activation of specificity protein 1 (Sp1) by AKT and Sp1 binding sites have been shown to be functional within the Sdf1 promoter (Gómez-Villafuertes et al. 2015; Pore et al. 2004; García-Moruja et al. 2005). Thus, Sdf1 expression might be modulated via Sp1 activated by AKT1.